# Learning under Model Misspecification: Applications to Variational and Ensemble methods

**Andrés R. Masegosa**[*]
University of Almería
andresma@ual.es

## Abstract

Virtually any model we use in machine learning to make predictions does not perfectly represent reality. So, most of the learning happens under model misspecification. In this work, we present a novel analysis of the generalization performance of Bayesian model averaging under model misspecification and i.i.d. data using a new family of second-order PAC-Bayes bounds. This analysis shows, in simple and intuitive terms, that Bayesian model averaging provides suboptimal generalization performance when the model is misspecified. In consequence, we provide strong theoretical arguments showing that Bayesian methods are not optimal for learning predictive models, unless the model class is perfectly specified. Using novel second-order PAC-Bayes bounds, we derive a new family of Bayesian-like algorithms, which can be implemented as variational and ensemble methods. The output of these algorithms is a new posterior distribution, different from the Bayesian posterior, which induces a posterior predictive distribution with better generalization performance. Experiments with Bayesian neural networks illustrate these findings.

## 1 Introduction

All our models are idealizations which only provide an approximation to the real-world distributions generating the data (i.e. "all models are wrong" [9]). But whether our models are or not well-specified is a key consideration in Bayesian statistics. Suboptimal behaviors of Bayesian methods when the model family is misspecified have been widely reported in the literature [20, 21, 22, 26, 51], even questioning the principles of Bayesian statistics.

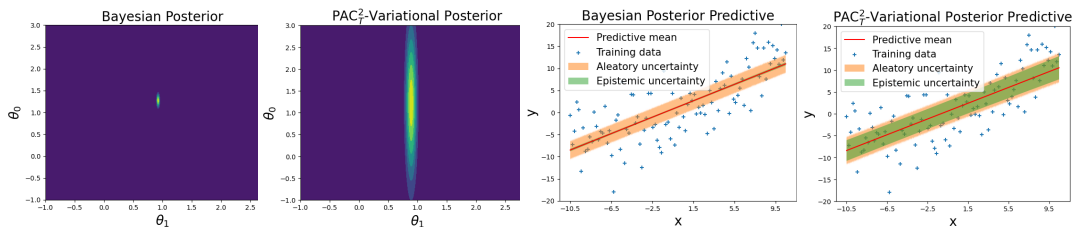

Figure 1: The exact Bayesian posterior and our new proposed (PAC$_T^2$-Variational) posterior, and their respective posterior predictive distributions, for a linear regression model with a misspecified constant noise term (the data noise is higher than the linear model's noise). The Bayesian posterior concentrates around the best single linear model, while our method estimates a posterior which introduces high variance in the intercept parameter $\theta_0$ to induce a posterior predictive distribution with higher noise that better fits the data distribution (see Appendix C.2 for details).

---

[*]Part of the work was done while AM was visiting the University of Copenhagen.

The use of Bayesian methods in machine learning (see [5, 19] for an introduction) has been in general very successful, specially for discovering hidden patterns in the data by inspecting the Bayesian posterior [8, 39, 50]. And model misspecification has not been considered as an unresolved issue [36, 38, 53, 54]. The focus has been put on approximate inference methods [23, 45].

At the same time, many other works have shown that Bayesian methods are not superior methods when the sole purpose is to make predictions (not to identify the true unknown parameters of a model) [10, 15, 17]. Bayesian methods make predictions through Bayesian model averaging, which combines the predictions of the individual models of the family weighted by their posterior probability. Ensemble models (see [12] for an introduction) are an alternative approach for model combination that have consistently provided very competitive generalization performance in a wide range of different problems, even in terms of well-calibrated probability predictions [49]. Recently, [56] provided strong evidence on how the generalization performance of Bayesian neural networks can be significantly improved by considering different posteriors distributions for model averaging that largely deviate from the Bayesian posterior.

**Contributions:** This paper provides a novel theoretical analysis of the generalization properties of Bayesian model averaging when the model family is misspecified. Our analysis shows that Bayesian model averaging provides suboptimal generalization performance because the Bayesian posterior is the minimum of a first-order PAC-Bayes bound [42], which can be quite loose when the model family is misspecified. Based on this analysis, we introduce novel second-order PAC-Bayes bounds and, based on the minimization of these bounds, we derive a new sound and scalable family of Bayesian-like algorithms with better generalization properties. These new algorithms can be interpreted as generalized variational methods [29] and, even, as ensemble methods. The output of these algorithms is a new posterior distribution, different from the Bayesian posterior, which induces new posterior predictive distributions with better generalization capacity. See Figure 1 for an illustrative example. Experiments with Bayesian neural networks also illustrate these findings.

## 2   Relevant prior work

PAC-Bayesian theory [42] provides probably approximately correct (PAC) bounds on the generalization risk (i.e., with probability $1 - \xi$, the generalization risk is at most $\epsilon$ away from the training risk.) Although PAC-Bayesian theory is mostly a *frequentist* method, connections between PAC-Bayes and Bayesian methods have been explored since the beginnings of the theory [33, 46]. But it was in [18] were a neat connection was established between Bayesian learning and PAC-Bayesian theory. However, they did not directly study the generalization performance of Bayesian model averaging and did not consider model misspecification.

There is a large literature showing that Bayesian inference can behave suboptimally if the model is wrong [20, 21, 22, 26, 51]. The *Safe Bayesian* method is probably the best-known framework [20]. The main point of this approach is to guarantee the concentration of the Bayesian posterior around the best possible model. But this work shows that the concentration of the Bayesian posterior around the best possible model is the main reason behind the suboptimal generalization performance of Bayesian methods under model misspecification.

Other related works [6, 11, 29, 34] propose Bayesian-like algorithms based on the use of alternative belief updating schemes which differs from the Bayesian approach. Again, the final goal of these works is not to study the generalization risk of Bayesian model averaging. Some of them [6, 29] are based on the use of alternative loss functions, different from the log-likelihood function, to derive new Bayesian-like algorithms. In this sense, our proposed learning algorithms employ a special loss which includes a correcting term to account for model misspecification.

Direct loss minimization [25, 48, 55] is a line of research close to our approach. These works analyze Bayesian methods from the angle of regularized loss minimization. They also consider the direct optimization of the log-loss of the posterior predictive distribution. But they do not consider a generalization performance analysis and the role that model misspecification has when justifying this approach with respect to standard Bayesian methods.

Zhang [60, 61] introduces information theoretical bounds which consider the log-loss and model misspecification. But the bounded quantity is not the generalization error of Bayesian model averaging, and their focus is to find the best single model, not the best model averaging distribution.

Robust Bayesian methods [4, 24, 28, 52, 54] also address the problem of model misspecification. But their focus is mainly in how to *fix* the inference procedure under *small deviations from the assumptions* (e.g. outliers, error measurements, etc) rather than systematically study the generalization performance under these circumstances.

## 3   Background

Our analysis is made under unsupervised settings or density estimation, but it readily applies to supervised classification settings too by considering labelled data and conditional probability distributions.

We denote the training data set $D = \{\boldsymbol{x}_1, \ldots, \boldsymbol{x}_n\}$, where $\boldsymbol{x} \in \mathcal{X}$. And the probability distribution over $\mathcal{X}$ is denoted by $p(\boldsymbol{x}|\boldsymbol{\theta})$, which is indexed by a parameter $\boldsymbol{\theta} \in \boldsymbol{\Theta}$. As a standard requirement in generalization analysis methods [42, 60], we assume that the samples in $D$ are i.i.d. generated from some unknown data generating distribution denoted $\nu(\boldsymbol{x})^2$. For this analysis, we assume that

**Assumption 1.** *There exists a constant $M < \infty$ such that $\forall \boldsymbol{x} \in \mathcal{X}, \forall \boldsymbol{\theta} \in \boldsymbol{\Theta}$, $p(\boldsymbol{x}|\boldsymbol{\theta}) \leq M$.*

This assumptions is always satisfied in supervised classifications settings (i.e, in this case we would have a conditional distribution $p(\boldsymbol{y}|\boldsymbol{x}, \boldsymbol{\theta})$ whose maximum is equal to 1). But, for example, when $p(\boldsymbol{x}|\boldsymbol{\theta})$ is a Normal density function, we have to restrict the parameter space $\boldsymbol{\Theta}$ to only consider variances higher than a given $\epsilon > 0$. Finally, we also assume that

**Assumption 2.** *We are learning under model misspecification, i.e., $\forall \boldsymbol{\theta} \in \boldsymbol{\Theta}$ $p(\cdot|\boldsymbol{\theta}) \neq \nu$.*

We define the *posterior predictive distribution* induced by a probability distribution $\rho$ over $\boldsymbol{\Theta}$ as $p(\boldsymbol{x}) = \mathbb{E}_{\rho(\boldsymbol{\theta})}[p(\boldsymbol{x}|\boldsymbol{\theta})]$, where $\rho$ is also referred as a posterior distribution because it depends on the data. When $\rho(\boldsymbol{\theta})$ is the Bayesian posterior, $\mathbb{E}_{\rho(\boldsymbol{\theta})}[p(\boldsymbol{x}|\boldsymbol{\theta})]$ corresponds to Bayesian model averaging.

We denote $CE(\rho)$ as the cross entropy of $\mathbb{E}_{\rho(\boldsymbol{\theta})}[p(\boldsymbol{x}|\boldsymbol{\theta})]$ wrt to $\nu(\boldsymbol{x})$,

$$CE(\rho) = \mathbb{E}_{\nu(\boldsymbol{x})}[-\ln \mathbb{E}_{\rho(\boldsymbol{\theta})}[p(\boldsymbol{x}|\boldsymbol{\theta})]]. \tag{1}$$

We address the problem of finding the optimal distribution $\rho^\star$ over $\boldsymbol{\Theta}$ for performing *model averaging*, in terms of generalization performance. So, we aim to find the probability distribution $\rho^\star$ which defines the *posterior predictive distribution* $p(\boldsymbol{x})$ with the smallest cross entropy wrt to the true data generating distribution $\nu(\boldsymbol{x})$, i.e., $\rho^\star = \arg\min_\rho CE(\rho)$. The distribution $\rho^\star$ also satisfies that $\rho^\star = \arg\min_\rho KL(\nu(\boldsymbol{x}), \mathbb{E}_{\rho(\boldsymbol{\theta})}[p(\boldsymbol{x}|\boldsymbol{\theta})])$, where $KL$ denotes the Kullback-Leibler (KL) divergence.

As the true distribution $\nu(\boldsymbol{x})$ is unknown, our approach to find $\rho^\star$ will be based on the minimization of a PAC-Bayes upper-bound [42], which depends on the data sample $D$, over the $CE(\rho)$ function. Note, $CE(\rho)$ is the expected log-loss of the posterior predictive distribution $\mathbb{E}_{\rho(\boldsymbol{\theta})}[p(\boldsymbol{x}|\boldsymbol{\theta})]$ and, in consequence, measures the generalization error associated to the density $\rho$.

### 3.1   Bayesian Learning and Variational Inference

The key quantity in *Bayesian statics* is the *Bayesian posterior*, $p(\boldsymbol{\theta}|D) \propto \pi(\boldsymbol{\theta}) \prod_{i=1}^n p(\boldsymbol{x}_i|\boldsymbol{\theta})$, where $\pi(\boldsymbol{\theta})$ is known as the *prior* distribution. When a new observation $\boldsymbol{x}'$ arrives we compute the *Bayesian posterior predictive* distribution to make predictions about $\boldsymbol{x}'$, $p(\boldsymbol{x}'|D) = \mathbb{E}_{p(\boldsymbol{\theta}|D)}[p(\boldsymbol{x}'|\boldsymbol{\theta})]$.

Variational Inference (VI) (see [7] for an introduction) is a popular method to compute approximations of intractable Bayesian posteriors. In standard VI settings, we choose a *tractable* family of probability distributions over $\boldsymbol{\Theta}$, denoted by $\mathcal{Q}$, and the learning problem consists in finding the probability distribution $q \in \mathcal{Q}$ which is closest to the Bayesian posterior in terms of the inverse KL divergence, $\arg\min_{q \in \mathcal{Q}} KL(q(\boldsymbol{\theta}), p(\boldsymbol{\theta}|D))$. Solving this minimization problem is equivalent to maximize the following function, which is known as the ELBO function,

$$q^\star(\boldsymbol{\theta}) = arg \max_{q \in \mathcal{Q}} \mathbb{E}_{q(\boldsymbol{\theta})}[\ln p(D|\boldsymbol{\theta})] - KL(q, \pi). \tag{2}$$

See [40, 59] for a recent review of methods to efficiently solve this maximization problem.

## 3.2 PAC-Bayesian Theory and Bayesian statistics

The PAC-Bayes framework [42] provides data-dependent generalization guarantees over the generalization error of a model under i.i.d. data. Let us define the expected log-loss for the model $\boldsymbol{\theta}$, denoted $L(\boldsymbol{\theta}) = \mathbb{E}_{\nu(\boldsymbol{x})}[-\ln p(\boldsymbol{x}|\boldsymbol{\theta})]$, and the empirical log-loss for the model $\boldsymbol{\theta}$ on the sample $D$, denoted $\hat{L}(\boldsymbol{\theta}, D) = \frac{1}{n}\sum_{i=1}^{n} -\ln p(\boldsymbol{x}_i|\boldsymbol{\theta})$. PAC-Bayesian theory provides probabilistic bounds over $\mathbb{E}_{\rho(\boldsymbol{\theta})}[L(\boldsymbol{\theta})]$ using $\mathbb{E}_{\rho(\boldsymbol{\theta})}[\hat{L}(\boldsymbol{\theta}, D)]$. But most of the PAC-Bayes bounds only apply to bounded losses and do not cover the log-loss, which is unbounded. [2] introduced a PAC-Bayes bound for a restrictive set of unbounded losses, which was later extended to general unbounded losses by [18, 47]. We reproduce here this PAC-Bayes bound [3] and, for completeness, a proof is given in Appendix A.1.

**Theorem 1.** *[18, 47] For any prior distribution $\pi$ over $\boldsymbol{\Theta}$ independent of $D$ and for any $\xi \in (0, 1)$ and $c > 0$, with probability at least $1 - \xi$ over draws of training data $D \sim \nu^n(\boldsymbol{x})$, for all distribution $\rho$ over $\boldsymbol{\Theta}$, simultaneously,*

$$\mathbb{E}_{\rho(\boldsymbol{\theta})}[L(\boldsymbol{\theta})] \leq \mathbb{E}_{\rho(\boldsymbol{\theta})}[\hat{L}(\boldsymbol{\theta}, D)] + \frac{KL(\rho, \pi) + \ln\frac{1}{\xi} + \psi_{\pi,\nu}(c, n)}{c\, n},$$

*where $\psi_{\pi,\nu}(c, n) = \ln \mathbb{E}_{\pi(\boldsymbol{\theta})}\mathbb{E}_{D\sim\nu^n(\boldsymbol{x})}[e^{c\, n(L(\boldsymbol{\theta}) - \hat{L}(\boldsymbol{\theta},D))}]$.*

But PAC-Bayes bounds also provide a well-founded approach to learning. As these bounds hold simultaneously for all densities $\rho$, the learning algorithm consists in choosing the distribution $\rho$ which minimizes the upper bound over the *generalization risk*. Fortunately, we can compute the $\rho$ distribution minimizing the PAC-Bayes bound of Theorem 1 for constant $c$, $\xi$, $n$ and $D$ values, because the $\psi_{\pi,\nu}(c, n)$ term is also constant wrt $\rho$. [18] noted that the Bayesian posterior distribution is the minimum of this PAC-Bayes bound over the expected log-loss $\mathbb{E}_{\rho(\boldsymbol{\theta})}[L(\boldsymbol{\theta})]$,

**Lemma 1.** *[18] The Bayesian posterior $p(\boldsymbol{\theta}|D)$ is the distribution over $\boldsymbol{\Theta}$ which minimizes the PAC-Bayes bound introduced in Theorem 1 for $c = 1$ and constant $\xi$, $n$ and $D$ values.*

## 4 The Bayesian posterior is suboptimal for generalization

In the previous section, we saw that the Bayesian posterior minimizes a PAC-Bayes upper bound over the expected log-loss. So, by minimizing the PAC-Bayes bound, we aim to minimize the expected log-loss $\mathbb{E}_{\rho(\boldsymbol{\theta})}[L(\boldsymbol{\theta})]$. In fact, under some technical conditions, the distribution minimizing the PAC-Bayes bound (i.e., the Bayesian posterior as shown in Lemma 1) converges, in the large sample limit and in probability, to a distribution minimizing the expected log-loss, $\mathbb{E}_{\rho(\boldsymbol{\theta})}[L(\boldsymbol{\theta})]$, due to well-known asymptotic results of the Bayesian posterior under model misspecification [27]. And this distribution can be characterized as a Dirac-delta distribution, denoted $\delta_{\boldsymbol{\theta}_{ML}^\star}(\boldsymbol{\theta})$, centered around $\boldsymbol{\theta}_{ML}^\star$, which is the parameter that minimizes the KL divergence wrt the true distribution, $\boldsymbol{\theta}_{ML}^\star = \arg\min_\theta KL(\nu(\boldsymbol{x}), p(\boldsymbol{x}|\boldsymbol{\theta}))$. This also applies for the variational posterior [53], i.e the variational posterior also converges in the large sample limit to $\delta_{\boldsymbol{\theta}_{ML}^\star}(\boldsymbol{\theta})$, a minimum of the expected log-loss. See Appendix A.3 for a formal proof of these statements.

But the question is whether the minimization of the expected log-loss, $\mathbb{E}_{\rho(\boldsymbol{\theta})}[L(\boldsymbol{\theta})]$, is a good strategy for minimizing the cross-entropy loss, $CE(\rho)$. In principle, this is a good strategy because, by the Jensen inequality, the expected log-loss is an upper oracle bound[4] of the cross-entropy loss,

$$\underbrace{\mathbb{E}_{\nu(\boldsymbol{x})}[-\ln\mathbb{E}_{\rho(\boldsymbol{\theta})}[p(\boldsymbol{x}|\boldsymbol{\theta})]]}_{CE(\rho)} \leq \underbrace{\mathbb{E}_{\rho(\boldsymbol{\theta})}[\mathbb{E}_{\nu(\boldsymbol{x})}[-\ln p(\boldsymbol{x}|\boldsymbol{\theta})]]}_{\mathbb{E}_{\rho(\boldsymbol{\theta})}[L(\boldsymbol{\theta})]}. \tag{3}$$

This strategy would be *optimal* if the minimum of the expected log-loss was also the minimum of the cross-entropy loss. But, as shown in the following result, this only happens when the best model in isolation, $p(\boldsymbol{x}|\boldsymbol{\theta}_{ML}^\star)$, provides better performance than any model averaging, $\mathbb{E}_{\rho(\boldsymbol{\theta})}[p(\boldsymbol{x}|\boldsymbol{\theta})]$,

**Lemma 2.** *A distribution minimizing $\mathbb{E}_{\rho(\boldsymbol{\theta})}[L(\boldsymbol{\theta})]$, denoted $\rho_{ML}^\star$, is also a minimizer of the cross-entropy loss $CE(\rho)$ if and only if for any distribution $\rho$ over $\boldsymbol{\Theta}$ we have that,*

$$KL(\nu(\boldsymbol{x}), p(\boldsymbol{x}|\boldsymbol{\theta}_{ML}^\star)) \leq KL(\nu(\boldsymbol{x}), \mathbb{E}_{\rho(\boldsymbol{\theta})}[p(\boldsymbol{x}|\boldsymbol{\theta})]).$$

And $\rho_{ML}^{\star}$ can always be characterized as a Dirac-delta distribution center around $\boldsymbol{\theta}_{ML}^{\star}$, i.e., $\rho_{ML}^{\star}(\boldsymbol{\theta}) = \delta_{\boldsymbol{\theta}_{ML}^{\star}}(\boldsymbol{\theta})$. *[Full proof in Appendix A.4].*

According to this result, the Bayesian posterior is an optimal learning strategy under perfect model specification because we have that $KL(\nu(\boldsymbol{x}), p(\boldsymbol{x}|\boldsymbol{\theta}_{ML}^{\star})) = 0$, and $\rho_{ML}^{\star}$ will be a minimum of $CE(\rho)$. But perfect model specification rarely happens in practice. The problem with the Bayesian posterior lies in the inequality of Equation (3), which is the result of the application of a first-order Jensen inequality [35]. And a first-order Jensen inequality induces a *linear* bound whose minimum is always at the border of the solution space, i.e., a Dirac-delta distribution. For this reason, we also refer to the expected log-loss $\mathbb{E}_{\rho(\boldsymbol{\theta})}[L(\boldsymbol{\theta})]$ as a first-order oracle bound, and to the PAC-Bayes bound of Theorem 1 as a first-order PAC-Bayes bound. But if we use a tighter second-order Jensen inequality [3, 35] to upper bound the cross-entropy loss, we will never end up in these extreme, no-model-averaging, solutions. Figure 2 graphically illustrates this situation.

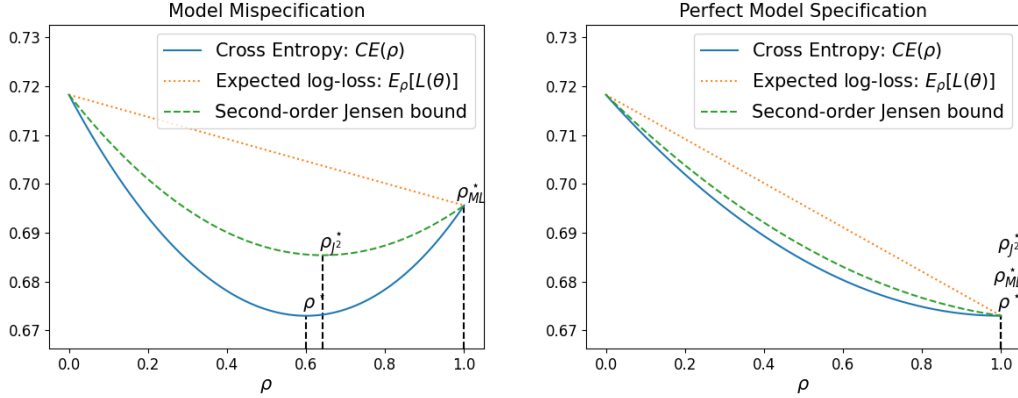

Figure 2: First-Order vs Second-Order Jensen Bounds. See Appendix B for full details.

## 5   Second-order PAC-Bayes bounds

We exploit second-order Jensen inequalities [3, 35] to derive tighter oracle bounds over $CE(\rho)$,

**Theorem 2.** *(Second-order Oracle bound) Any distribution $\rho$ over $\boldsymbol{\Theta}$ satisfies that,*

$$CE(\rho) \leq \mathbb{E}_{\rho(\boldsymbol{\theta})}[L(\boldsymbol{\theta})] - \mathbb{V}(\rho),$$

*where $\mathbb{V}(\rho)$ is a variance term defined as*

$$\mathbb{V}(\rho) = \mathbb{E}_{\nu(\boldsymbol{x})}\left[\frac{1}{2\max_{\boldsymbol{\theta}} p(\boldsymbol{x}|\boldsymbol{\theta})^2}\mathbb{E}_{\rho(\boldsymbol{\theta})}[(p(\boldsymbol{x}|\boldsymbol{\theta}) - p(\boldsymbol{x}))^2]\right],$$

*where $\max_{\boldsymbol{\theta}} p(\boldsymbol{x}|\boldsymbol{\theta})^2$ is a finite scalar value according to Assumption 1.*

*Proof sketch.* Apply [3, 35] to the random variable $p(\boldsymbol{x}|\boldsymbol{\theta})$. Full proof in Appendix A.5. $\qquad\square$

This second-order oracle bound differs from the expected log-loss, $\mathbb{E}_{\rho(\boldsymbol{\theta})}[L(\boldsymbol{\theta})]$, (a first-order oracle bound) in this new variance term $\mathbb{V}(\rho)$, which is positive when the $\rho$ distribution is not a Dirac-delta distribution. So, this second-order oracle bound is tighter than the the expected log-loss and, also, induces high variance solutions when it is minimized.

But the key point is that a distribution minimizing this new second-order oracle bound induces better model averaging solutions than a distribution minimizing the expected log-loss,

**Lemma 3.** *Let us denote $\rho_{J^2}^{\star}$ and $\rho_{ML}^{\star}$ a distribution minimizing the second-order oracle bound of Theorem 2 and $\mathbb{E}_{\rho(\boldsymbol{\theta})}[L(\boldsymbol{\theta})]$, respectively. The following inequality holds*

$$KL(\nu(\boldsymbol{x}), \mathbb{E}_{\rho_{J^2}^{\star}}[p(\boldsymbol{x}|\boldsymbol{\theta})]) \leq KL(\nu(\boldsymbol{x}), \mathbb{E}_{\rho_{ML}^{\star}}[p(\boldsymbol{x}|\boldsymbol{\theta})]),$$

*and the equality holds if we are under perfect model specification.*

A proof is provided in Appendix A.6. The above result shows that a learning strategy based on the minimization of this second-order oracle bound is better than a learning strategy based on the minimization of the expected log-loss (which is the strategy of Bayesian learning). In fact, in case of perfect model specification, the minimum of the second-order oracle bound equals the minimum of the expected log-loss.

However, the direct minimization of the second-order oracle bound of Theorem 2 is not possible because it assumes access to $\nu(\boldsymbol{x})$. But the following result introduces a second-order PAC-Bayes bound over the second-order oracle bound, which also provides generalization guarantees over the performance of the posterior predictive distribution, because it also bounds the cross-entropy loss.

**Theorem 3.** *(Second-order PAC-Bayes bound) For any prior distribution $\rho$ over $\Theta$ independent of $D$ and for any $\xi \in (0,1)$ and $c > 0$, with probability at least $1 - \xi$ over draws of training data $D \sim \nu^n(\boldsymbol{x})$, for all distribution $\rho$ over $\Theta$, simultaneously,*

$$CE(\rho) \leq \mathbb{E}_{\rho(\boldsymbol{\theta})}[L(\boldsymbol{\theta})] - \mathbb{V}(\rho) \leq \mathbb{E}_{\rho(\boldsymbol{\theta})}[\hat{L}(\boldsymbol{\theta}, D)] - \hat{\mathbb{V}}(\rho, D) + \frac{KL(\rho, \pi) + \frac{1}{2}\ln\frac{1}{\xi} + \frac{1}{2}\psi'_{\pi, \nu}(c, n)}{cn},$$

*where $\psi'_{\pi, \nu}(c, n)$ is the same term as in Theorem 1 adapted to this setting and $\hat{\mathbb{V}}(\rho, D)$ is the empirical version of $\mathbb{V}(\rho)$.*

*Proof sketch.* We express the problem using a *tandem log-loss*. Note that $\mathbb{E}_{\rho(\boldsymbol{\theta})}[L(\boldsymbol{\theta})] - \mathbb{V}(\rho) = \mathbb{E}_{\boldsymbol{\theta} \sim \rho, \boldsymbol{\theta}' \sim \rho}[L(\boldsymbol{\theta}, \boldsymbol{\theta}')]$, where $L(\boldsymbol{\theta}, \boldsymbol{\theta}') = \mathbb{E}_{\nu(\boldsymbol{x})}[\ln\frac{1}{p(\boldsymbol{x}|\boldsymbol{\theta})} - \frac{1}{2\max_{\boldsymbol{\theta}} p(\boldsymbol{x}|\boldsymbol{\theta})^2}(p(\boldsymbol{x}|\boldsymbol{\theta})^2 - p(\boldsymbol{x}|\boldsymbol{\theta})p(\boldsymbol{x}|\boldsymbol{\theta}'))]$. Then, we apply [18, Theorem 3] to this loss and fix $\lambda = 2cn$. $\psi'_{\pi, \nu}(c, n)$ is like the $\psi_{\pi, \nu}(c, n)$ term of Theorem 1 adapted to $L(\boldsymbol{\theta}, \boldsymbol{\theta}')$. Full proof in Appendix A.7. □

As happen with the bound presented in Theorem 1, this bound can not be directly computed because the $\psi'_{\pi, \nu}(c, n)$ term depends on $\nu(\boldsymbol{x})$. We could apply the approaches presented in [1, 18] to provide computable upper bounds over $\psi'_{\pi, \nu}(c, n)$, but it would require strong assumptions and would only be applicable to very simple models. Fortunately, we can still minimize this PAC-Bayes bound for constant $c, \xi, n$ and $D$ values, because $\psi'_{\pi, \nu}(c, n)$ is also constant wrt $\rho$.

The key part of this new PAC-Bayes bound is the variance term $\mathbb{V}(\rho, D)$, which measures the *diversity* or the *disagreement* among the predictions of the models. Note, for example, that when all the models provide the same predictions the variance term is null and there is no gain in making model averaging with these models. *Diversity* or *disagreement* among models has been empirically identified as a key factor in the performance of model combination [14, 31]. This work describes which should be the precise balance between the average empirical log-loss $\mathbb{E}_{\rho(\boldsymbol{\theta})}[\hat{L}(\boldsymbol{\theta}, D)]$ of the models (i.e., how well the models individually fit the training data) and how difference they should be among them (i.e. measure through $\mathbb{V}(\rho, D)$) to maximize the generalization performance of the model averaging. A recent work [41] arrives at similar conclusions in the context of weighted majority voting.

# 6 Learning by minimizing second-order PAC-Bayes bounds

Our learning strategy is then to minimize the second-order PAC-Bayes bound introduced in Theorem 3 because it is a *probabilistic approximate correct* bound over the generalization error of the resulting posterior predictive distribution. In this case, we do not have a closed-form solution to find the distribution $\rho$ minimizing this second-order PAC-Bayes bound. But, in the next subsections, we introduce several scalable methods for (approximately) solving this minimization problem.

## 6.1 PAC$^2$-Variational Learning

Like in variational inference (see Section 3.1), we can choose a tractable family of densities $\rho(\boldsymbol{\theta}|\boldsymbol{\lambda}) \in \mathcal{Q}$, parametrized by some parameter vector $\boldsymbol{\lambda}$, to solve the minimization of the second-order PAC-Bayes bound of Theorem 3. By discarding constant terms of this bound wrt $\rho$ and setting $c = 1$ in order to keep the connection with Bayesian approaches[5], the minimization problem can be written as,

$$\arg\min_{\boldsymbol{\lambda}} \mathbb{E}_{\rho(\boldsymbol{\theta}|\boldsymbol{\lambda})}[\hat{L}(\boldsymbol{\theta}, D)] - \hat{\mathbb{V}}(\rho, D) + \frac{KL(\rho, \pi)}{n}. \tag{4}$$

We refer to this learning method as *PAC$^2$-Variational learning*, which can be interpreted as a generalized variational method [29]. Note, the standard variational inference algorithm (see Equation (2)) can be similarly derived by minimizing the PAC-Bayes bound of Theorem 1, which misses the $\hat{\mathbb{V}}(\rho, D)$ term because it is based on a first-order oracle bound. Appendix C.2 shows a numerically stable version of the *PAC$^2$-Variational learning* to perform optimization over this objective function using modern black-box variational methods [59].

## 6.2  PAC$^2_T$-Variational Learning

One of the key contributions of our work is to show that the error induced when bounding the cross-entropy loss is a significant barrier when learning under model misspecification. Our assumption is that our learning strategy should further improve if we use tighter second-order Jensen bounds. [35] proposed an alternative second-order Jensen bound which is tighter than the one considered in Theorem 2. This new bound suggests a new learning algorithm, referred as *PAC$^2_T$-Variational learning*, already illustrated in Figure 1 for a linear model. The subscript $T$ highlights that it relies on *tighter* Jensen bounds. The only difference with the approach presented in Equation (4) is the use of a different variance term, denoted $\hat{\mathbb{V}}_T(\rho, D)$,

$$\hat{\mathbb{V}}_T(\rho, D) = \frac{1}{n} \sum_{i=1}^{n} h(m_{\boldsymbol{x}_i}, \mu_{\boldsymbol{x}_i}) \mathbb{E}_{\rho(\boldsymbol{\theta})}[(p(\boldsymbol{x}_i|\boldsymbol{\theta}) - p(\boldsymbol{x}_i))^2], \tag{5}$$

where $\mu_{\boldsymbol{x}_i} = \mathbb{E}_{\rho(\boldsymbol{\theta})}[p(\boldsymbol{x}_i|\boldsymbol{\theta})]$, $m_{\boldsymbol{x}_i} = \max_{\boldsymbol{\theta}} p(\boldsymbol{x}_i|\boldsymbol{\theta})$ and $h(m, \mu) = \frac{\ln \mu - \ln m}{(m-\mu)^2} + \frac{1}{\mu(m-\mu)}$. We provide a formal proof for this new tighter bound in Appendix C.1. A numerically stable version of this learning algorithm is provided in Appendix C.2. Figures C.5, C.6, C.8 and C.9 illustrate the behavior of the two presented versions of the PAC$^2$-Variational learning algorithm in several toy examples.

## 6.3  PAC$^2$-Ensemble Learning

Ensemble models (see [12] for an introduction) are based on the combination of a finite set of models to obtain better predictions than the predictions of a single model alone. This section provides an adaptation of the previous results for learning a finite set of models (i.e., an ensemble model). As a consequence, we provide a novel explanation of why the so-called *diversity* of the ensemble [31] is key to have powerful ensembles. We also present a novel ensemble learning algorithm.

We first assume that $\boldsymbol{\Theta} \subseteq \mathcal{R}^M$. Let us denote $\rho_E$ a mixture of Dirac-delta distributions centered around a set of $E$ parameters $\{\boldsymbol{\theta}_j\}_{1 \leq j \leq E}$,

$$\rho_E(\boldsymbol{\theta}) = \sum_{j=1}^{E} \frac{1}{E} \delta_{\boldsymbol{\theta}_j}(\boldsymbol{\theta}).$$

So, we have that $\mathbb{E}_{\rho_E(\boldsymbol{\theta})}[p(\boldsymbol{x}|\boldsymbol{\theta})] = \frac{1}{E} \sum_{j=1}^{E} p(\boldsymbol{x}|\boldsymbol{\theta}_j)$, i.e. the averaging of a finite set of models.

In order to properly define the Kullback-Leibler divergence between $\rho_E$ and a given prior, we restrict ourselves to the following family of priors, denoted $\pi_F(\boldsymbol{\theta})$. For any prior $\pi_F(\boldsymbol{\theta})$ within this family, its support is contained in $\boldsymbol{\Theta}_F$, which denotes the space of real number vectors of dimension $M$ that can be represented under a finite-precision scheme using $F$ bits to encode each element of the vector. So, we have that $supp(\pi_F) \subseteq \boldsymbol{\Theta}_F \subseteq \boldsymbol{\Theta} \subseteq \mathcal{R}^M$. This prior distribution $\pi_F$ can be expressed as, $\pi_F(\boldsymbol{\theta}) = \sum_{\boldsymbol{\theta}' \in \boldsymbol{\Theta}_F} w_{\boldsymbol{\theta}'} \delta_{\boldsymbol{\theta}'}(\boldsymbol{\theta})$, where $w_{\boldsymbol{\theta}'}$ are positive scalars values parametrizing this prior distribution. They satisfy that $w_{\boldsymbol{\theta}'} \geq 0$ and $\sum w_{\boldsymbol{\theta}'} = 1$.

The following result provides a second-order PAC-Bayes bound for an ensemble of models,

**Theorem 4.** *For any prior distribution $\pi_F$ over $\boldsymbol{\Theta}_F$ and independent of $D$ and for any $\xi \in (0, 1)$ and $c > 0$, with probability at least $1 - \xi$ over draws of training data $D \sim \nu^n(\boldsymbol{x})$, for all densities $\rho_E$ with $supp(\rho_E) \subseteq \boldsymbol{\Theta}_F$, simultaneously,*

$$CE(\rho_E) \leq \mathbb{E}_{\rho_E(\boldsymbol{\theta})}[\hat{L}(\boldsymbol{\theta}, D)] - \hat{\mathbb{V}}(\rho_E, D) + \frac{KL(\rho_E, \pi_F) + \frac{1}{2}\ln\frac{1}{\xi} + \frac{1}{2}\psi'_{\pi_F, \nu}(c, n)}{cn},$$

*where $\psi'_{\pi_F, \nu}(c, n)$ is the same term as in Theorem 3, and $\hat{\mathbb{V}}(\rho_E, D)$ is the empirical variance.*

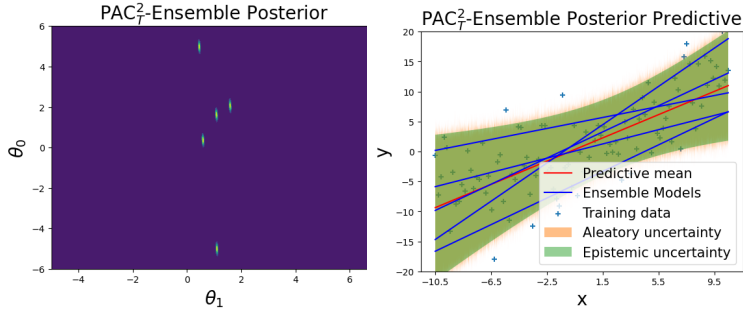

Figure 3: The PAC$_T^2$-Ensemble posterior, $\rho_E$, and its posterior predictive distribution for the same misspecified linear regression model used in Figure 1. The ensemble used 5 linear regression models which do not collapse and better approximates the data distribution (see Appendix C.3 for details).

A proof is provided in Appendix C.3. Note that $\hat{\mathbb{V}}(\rho_E, D)$ can be interpreted as a measure of the *diversity* of the ensemble [31]. According to the above result, to learn optimal ensembles, we need to trade-off how well we fit the data (i.e., low values for $\mathbb{E}_{\rho_E(\boldsymbol{\theta})}[\hat{L}(\boldsymbol{\theta}, D)]$) while also maintaining high *diversity* (i.e., high $\hat{\mathbb{V}}(\rho_E, D)$ values) to make the ensemble work (i.e., to reduce the generalization bound given in Theorem 4). This is a clear explanation of why ensembles need diversity to generalize better, but it is out the scope of the this paper to explore this claim.

We derive again a learning algorithm for building ensembles by minimizing the generalization upper bound of Theorem 4. Similarly to the variational methods, we derived two versions, the *PAC$^2$-Ensemble* and the *PAC$_T^2$-Ensemble* learning algorithms, based on the use of $\hat{\mathbb{V}}(\rho_E, D)$ and $\hat{\mathbb{V}}_T(\rho_E, D)$ (see Equation (5)), respectively. Note that these learning algorithms could be interpreted as particle-based variational inference methods [37], because the posterior $\rho_E$ is represented as a set of *particles*, which could be potentially very expressive. Figure 3 illustrates one of these algorithms for a simple linear regression model. In Appendix C.3, we provide all the details of these algorithms and illustrate their behavior on another toy models, including one with multimodal posteriors.

We can similarly derive an ensemble algorithm from the PAC-Bayes bound of Theorem 1 using $\rho_E$ and $\pi_E$ densities. In this case, the $\hat{\mathbb{V}}(\rho_E, D)$ term would disappear from the PAC-Bayes bound, and the algorithm will not induce diversity among the ensemble members. For example, this algorithm could recover as an optimal solution a collapsed ensemble with all models equal to the MAP model, which is equivalent to a single ensemble model. See Appendix C.3 for a formal proof of this statement.

In Appendix C.4, we also show how all the learning algorithms presented in Section 6, based on the minimization of second-order PAC-Bayes bounds, behave quite similarly to their first-order counterparts when the model family is not misspecified (see Lemma 3 and the subsequent discussion).

## 7   Empirical Evaluation

We performed the empirical evaluation on two data sets, Fashion-MNIST [58] and CIFAR-10 [30] and two prediction tasks [6]. A standard supervised task and a self-supervised task, where the goal is to predict the pixels of the below half part of the image given the pixels of the upper half part of the same image. For the self-supervised task, we employed two data models: a Normal distribution for continuous value predictions and a Binomial one for binarized pixels. The prediction model was a multi-layer perceptron with 20 hidden units. We always assumed the same standard normal prior and a fully factorized mean-field normal distribution. The generalization risk was evaluated by computing the average negative log-likelihood (NLL) on independent test sets. Full details in Appendix D.

Figure 4 shows the result of this evaluation. These results validate the main hypothesis of our work: the use of tighter bounds addressing the gap introduced when upper bounding the cross-entropy loss $CE(\rho)$ leads to learning algorithms that generalize better. PAC$^2$-Variational and PAC$^2$-Ensemble methods, based on second-order bounds, have better predictive performance than standard variational

methods and single model ensembles, which are based on first-order bounds (see the discussions at the end of Section 6.1 and Section 6.3, respectively). And, in turn, the $PAC_T^2$-Variational and $PAC_T^2$-Ensemble methods, based on tighter second-order bounds, generalize better than the $PAC^2$-Variational and $PAC^2$-Ensemble methods, respectively. The only exception is the classification task in CIFAR-10, where all the variational approaches do not perform better than the MAP model. In our opinion, this is due to poor prior specification, as discussed in [56], or because we employ a too-simplistic variational family for these settings.

Ensemble methods (see Section 6.3) clearly outperform over the rest. We argue this is mainly because the *variational family* $\rho_E$, based on mixtures of Dirac-delta distributions, is much more flexible than standard mean-field variational approximations. In fact, $\rho_E$ could even represent multimodal distributions (see Appendix C.3 for a concrete example).

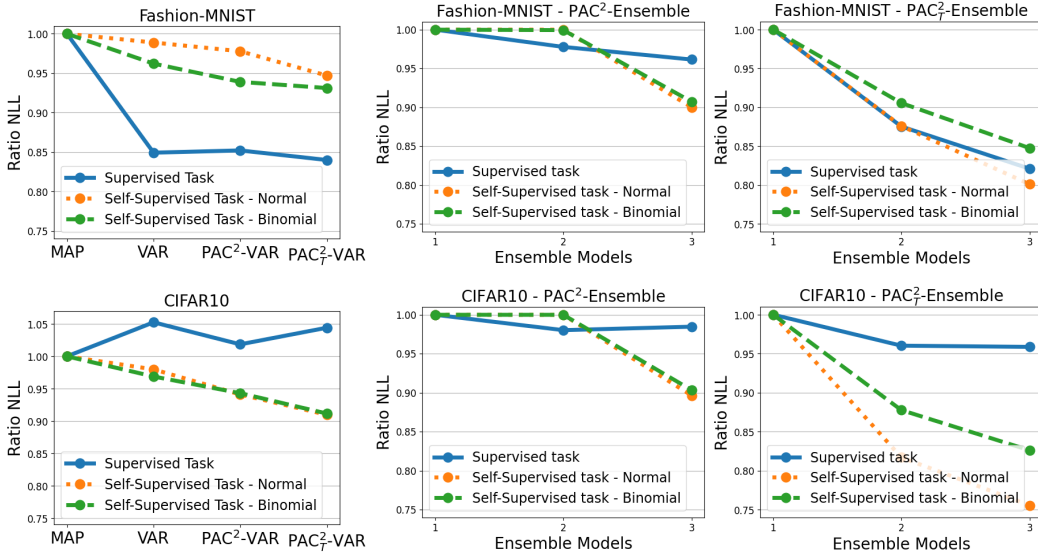

Figure 4: **Experimental Evaluation.** First column shows the ratio of the test negative log-likelihood (NLL) wrt to a MAP model for the Variational (VAR), $PAC^2$-Variational ($PAC^2$-VAR) and $PAC_T^2$-Variational ($PAC_T^2$-VAR) methods (e.g. ratio = 0.95 indicates a 5% improvement in NLL w.r.t. a MAP model). Second and third columns similarly evaluates $PAC^2$-Ensembles and $PAC_T^2$-Ensembles w.r.t. a MAP model (or, equivalently, an ensemble with a single model). All the models of the ensembles are randomly initialized with the same parameters and are jointly optimized with the same mini-batches, so it is the variance term $\hat{\mathbb{V}}(\rho_E, D)$ the only mechanism which induces better generalization performance, because, otherwise, all the models of the ensemble would be identical.

# 8 Discussion

This work performs a novel theoretical analysis of the generalization capacity of Bayesian model averaging under model misspecification and provides strong theoretical arguments showing that Bayesian methods are suboptimal for learning predictive models when the model family is misspecified.

These theoretical insights can be of help to better understand the generalization performance of Bayesian approaches. For example, in many cases, Bayesian neural networks do not outperform standard methods [44, 56]. Our work shows that, if a neural network does not perfectly represent reality, Bayesian learning methods do not provide optimal generalization performance.

This work may also help to better explain the relationship between ensembles and Bayesian approaches. Deep ensemble models [32, 49] provide SOTA performance for uncertainty estimation. [57] argues that ensembles are approximate Bayesian methods, which are able to capture multimodality. But we provide an alternative theoretical explanation. We show that we need to induce *diversity* [31], measured by the variance term of the second-order PAC-Bayes bound, to define ensembles of models that generalize. We hypothesise that the random initialization of each member of the ensemble is one of the key ingredients to make them diverse.

## Acknowledgments and Disclosure of Funding

We thank Yevgeny Seldin and the anonymous reviewers for their suggestions for manuscript improvements. This work is funded by the Spanish Ministry of Science, Innovation and Universities under the projects TIN2015-74368-JIN, TIN2016-77902-C3-3-P and PID2019-106758GB-C32, and by a Jose Castillejo scholarship CAS19/00279.

## Broader impact

Machine learning models are quickly playing a prominent role in society, industries, and individuals. In consequence, there is a growing demand to have machine learning models that can assert the confidence they have in their predictions, specially, to avoid catastrophic decisions. Predictive models which provide well-calibrated probabilities are a sound way to attach a confidence level to a prediction. Bayesian methods are the main tools employed for this goal. This work provides novel theoretical tools to better understand why Bayesian methods induce predictive models with suboptimal performance in terms of well-calibrated probabilities. So, the findings of this work can be of help to develop more accurate and safer predictive models in machine learning, which could ease the adoption of this technology.

## Footnotes

[2]We assume that $p(\cdot|\boldsymbol{\theta})$ and $\nu$ are probability measures having densities w.r.t. the Lebesgue measure.

[3]This bound is expressed in terms of any $\lambda > 0$, which we equivalently set here as $\lambda = c\, n$, for any $c > 0$.

[4]An oracle bound depends on the unknown distribution $\nu(\boldsymbol{x})$.

[5]Appendix C.5 further discusses how to set this parameter.

[6]The code to reproduce the results is available in `https://github.com/PGM-Lab/PAC2BAYES`.

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
