[Supplementary Material]

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

# A Proofs

## A.1 Proof of Theorem 1

**Theorem 1.** *[18, 47] For any prior distribution $\pi$ over $\Theta$ independent of $D$ and for any $\xi \in (0,1)$ and $c > 0$, with probability at least $1 - \xi$ over draws of training data $D \sim \nu^n(\boldsymbol{x})$, for all distribution $\rho$ over $\Theta$, simultaneously,*

$$\mathbb{E}_{\rho(\boldsymbol{\theta})}[L(\boldsymbol{\theta})] \leq \mathbb{E}_{\rho(\boldsymbol{\theta})}[\hat{L}(\boldsymbol{\theta}, D)] + \frac{KL(\rho, \pi) + \ln \frac{1}{\xi} + \psi_{\pi, \nu}(c, n)}{c\, n},$$

*where $\psi_{\pi, \nu}(c, n) = \ln \mathbb{E}_{\pi(\boldsymbol{\theta})} \mathbb{E}_{D \sim \nu^n(\boldsymbol{x})}[e^{c\, n(L(\boldsymbol{\theta}) - \hat{L}(\boldsymbol{\theta}, D))}]$.*

*Proof.* The Donsker-Varadhan's change of measure theorem states that for any measurable function $\phi : \Theta \to \Re$, we have $\mathbb{E}_{\rho(\boldsymbol{\theta})}[\phi(\boldsymbol{\theta})] \leq KL(\rho, \pi) + \ln \mathbb{E}_{\pi(\boldsymbol{\theta})}[e^{\phi(\boldsymbol{\theta})}]$. Thus, with $\phi(\boldsymbol{\theta}) = \lambda(L(\boldsymbol{\theta}) - \hat{L}(\boldsymbol{\theta}, D))$, we have that for any distribution $\rho$ over $\Theta$,

$$\mathbb{E}_{\rho(\boldsymbol{\theta})}[\lambda(L(\boldsymbol{\theta}) - \hat{L}(\boldsymbol{\theta}, D))] \leq KL(\rho, \pi) + \ln \mathbb{E}_{\pi(\boldsymbol{\theta})}[e^{\lambda(L(\boldsymbol{\theta}) - \hat{L}(\boldsymbol{\theta}, D))}] \tag{A.6}$$

Let us consider the non-negative random variable $\zeta = \mathbb{E}_{\pi(\boldsymbol{\theta})}[e^{\lambda(L(\boldsymbol{\theta}) - \hat{L}(\boldsymbol{\theta}, D))}]$. By the Markov inequality we have,

$$\mathbb{P}(\zeta \leq \frac{1}{\xi} \mathbb{E}_D[\zeta]) \geq 1 - \xi$$

which, combined with Equation (A.6), gives

$$\mathbb{P}\left(\mathbb{E}_{\rho(\boldsymbol{\theta})}[\lambda(L(\boldsymbol{\theta}) - \hat{L}(\boldsymbol{\theta}, D))] \leq KL(\rho, \pi) + \ln \frac{1}{\xi} \mathbb{E}_D[\zeta]\right) \geq 1 - \xi$$

By rearranging the terms of above equation, we obtain the following equivalent form of the statement of the theorem,

$$\mathbb{P}\left(\mathbb{E}_{\rho(\boldsymbol{\theta})}[L(\boldsymbol{\theta})] \leq \mathbb{E}_{\rho(\boldsymbol{\theta})}[\hat{L}(\boldsymbol{\theta}, D)] + \frac{KL(\rho, \pi) + \ln \frac{1}{\xi} + \ln \mathbb{E}_D[\zeta]}{\lambda}\right) \geq 1 - \xi$$

Finally, by setting $\lambda = c\, n$, we can see that $\ln \mathbb{E}_D[\zeta] = \psi_{\pi, \nu}(c, n)$, and we prove the statement of the theorem. $\qquad \square$

## A.2 The Dirac-delta function

The Dirac-delta function [16], $\delta_{\boldsymbol{\theta}_0} : \Theta \to \mathcal{R}^+$ with parameter $\boldsymbol{\theta}_0 \in \Theta$, is a generalized function with the following property

$$\mathbb{E}_{\delta_{\boldsymbol{\theta}_0}(\boldsymbol{\theta})}[f(\boldsymbol{\theta})] = \int \delta_{\boldsymbol{\theta}_0}(\boldsymbol{\theta}) f(\boldsymbol{\theta}) d\boldsymbol{\theta} = f(\boldsymbol{\theta}_0) \tag{A.7}$$

for any continuous function $f : \Theta \to \mathcal{R}$. The Dirac-delta function also defines the density function of a probability distribution because it is positive and, by the above property, $\int \delta_{\boldsymbol{\theta}_0}(\boldsymbol{\theta}) d\boldsymbol{\theta} = 1$. This Dirac-delta distribution is a degenerated probability distribution which always samples the same value $\boldsymbol{\theta}_0$ because $\delta_{\boldsymbol{\theta}_0}(\boldsymbol{\theta}) = 0$ if $\boldsymbol{\theta} \neq \boldsymbol{\theta}_0$ [16].

## A.3 Minimum of the expected log-loss

The following lemma defines that a Dirac-delta distribution center around $\boldsymbol{\theta}^\star_{ML}$ is a minimum of the expected log-loss,

**Lemma A.4.** *The distribution $\rho^\star_{ML}$ defined as a Dirac-delta distribution center around $\boldsymbol{\theta}^\star_{ML}$, $\rho^\star_{ML}(\boldsymbol{\theta}) = \delta_{\boldsymbol{\theta}^\star_{ML}}(\boldsymbol{\theta})$ is a minimum of the expected log-loss,*

$$\rho^\star_{ML} = \arg \min_\rho \mathbb{E}_{\rho(\boldsymbol{\theta})}[L(\boldsymbol{\theta})],$$

*where $\boldsymbol{\theta}^\star_{ML} = \arg \min_\theta KL(\nu(\boldsymbol{x}), p(\boldsymbol{x}|\boldsymbol{\theta}))$.*

*Proof.* We first have that $KL(\nu(\boldsymbol{x}), p(\boldsymbol{x}|\boldsymbol{\theta})) = L(\boldsymbol{\theta}) - H(\nu)$, where $H(\nu)$ denotes the entropy of $\nu(\boldsymbol{x})$. As $H(\nu)$ is constant wrt $\boldsymbol{\theta}$, $\boldsymbol{\theta}^\star_{ML}$ is also a minimum of $L(\boldsymbol{\theta})$. We also have that $\int (L(\boldsymbol{\theta}) - H(\nu))\rho(\boldsymbol{\theta})d\boldsymbol{\theta} \geq \min_{\boldsymbol{\theta}}(L(\boldsymbol{\theta}) - H(\nu)) \int \rho(\boldsymbol{\theta})d\boldsymbol{\theta}$, because $L(\boldsymbol{\theta}) - H(\nu) \geq 0$. Rearranging terms, we have that $\mathbb{E}_{\rho(\boldsymbol{\theta})}[L(\boldsymbol{\theta})] \geq L(\boldsymbol{\theta}^\star_{ML}) = \mathbb{E}_{\delta_{\boldsymbol{\theta}^\star_{ML}}(\boldsymbol{\theta})}[L(\boldsymbol{\theta})]$, where the last equality follows from standard properties of Dirac-delta distributions (see Appendix A.2). $\qquad\square$

This result states that both the Bayesian and the Variational posterior converge to a minimum of the expected log-loss,

**Lemma A.5.** *Under the technical conditions established in [27, 53], the Bayesian posterior $p(\boldsymbol{\theta}|D)$ and the Variational posterior $q^\star(\boldsymbol{\theta})$ converge, in the large sample limit and in probability, to a minimum of the expected log-loss, $\mathbb{E}_{\rho(\boldsymbol{\theta})}[L(\boldsymbol{\theta})]$.*

*Proof.* This results follows from the convergence results given in [27, 53], which state that $p(\boldsymbol{\theta}|D)$ and $q^\star(\boldsymbol{\theta})$ converge, in the large sample limit and in probability, to $\delta_{\boldsymbol{\theta}^\star_{ML}}(\boldsymbol{\theta})$. And, by Lemma A.4, $\delta_{\boldsymbol{\theta}^\star_{ML}}(\boldsymbol{\theta})$ is a minimum of the expected log-loss. $\qquad\square$

We now characterize any distribution minimizing the expected log-loss,

**Lemma A.6.** *Let us denote $\Omega_{\boldsymbol{\theta}_0}$ the set of parameters which defines the same distribution than $\boldsymbol{\theta}_0$, i.e. if $\boldsymbol{\theta}' \in \Omega_{\boldsymbol{\theta}_0} \subseteq \Theta$ then $\forall \boldsymbol{x} \in supp(\nu) \subseteq \mathcal{X}$ $p(\boldsymbol{x}|\boldsymbol{\theta}_0) = p(\boldsymbol{x}|\boldsymbol{\theta}')$, where $supp(\nu)$ denotes the support of $\nu(\boldsymbol{x})$. We have that any distribution $\rho^\star$ which is a minimum of the expected log-loss, $\rho^\star = \arg\min_\rho \mathbb{E}_{\rho(\boldsymbol{\theta})}[L(\boldsymbol{\theta})]$, satisfies that $supp(\rho^\star) \subseteq \Omega_{\boldsymbol{\theta}^\star_{ML}}$, where $supp(\rho^\star)$ denotes the support of $\rho^\star(\boldsymbol{\theta})$, and also that $\mathbb{V}(\rho^\star) = 0$.*

*Proof.* From Lemma A.4, we have that $\rho^\star_{ML}(\boldsymbol{\theta}) = \delta_{\boldsymbol{\theta}^\star_{ML}}(\boldsymbol{\theta})$ is a minimum of the expected log-loss. In consequence, if $\rho^\star$ is a minimum of the expected log-loss, then $\mathbb{E}_{\rho^\star}[L(\boldsymbol{\theta})] = \mathbb{E}_{\rho^\star_{ML}}[L(\boldsymbol{\theta})]$. From this equality, we arrive to the following equality $\mathbb{E}_{\rho^\star}[L(\boldsymbol{\theta}) - L(\boldsymbol{\theta}^\star_{ML})] = 0$. And this last equality can only be satisfied if the conditions of the lemma hold because, as noted in the proof of Lemma A.4, $\boldsymbol{\theta}^\star_{ML}$ is a minimum of $L(\boldsymbol{\theta})$ and, as a consequence, $L(\boldsymbol{\theta}) - L(\boldsymbol{\theta}^\star_{ML}) \geq 0$.

As $supp(\rho^\star) \subseteq \Omega_{\boldsymbol{\theta}^\star_{ML}}$, we have that, by definition, $\forall \boldsymbol{\theta}' \in supp(\rho^\star), \forall \boldsymbol{x} \in supp(\nu) \subseteq \mathcal{X}$ $p(\boldsymbol{x}|\boldsymbol{\theta}^\star_{ML}) = p(\boldsymbol{x}|\boldsymbol{\theta}')$. And we can deduce that $\mathbb{V}(\rho^\star) = 0$. $\qquad\square$

## A.4 Proof of Lemma 2

**Lemma 2.** *A distribution minimizing $\mathbb{E}_{\rho(\boldsymbol{\theta})}[L(\boldsymbol{\theta})]$, denoted $\rho^\star_{ML}$, is also a minimizer of the cross-entropy loss $CE(\rho)$ if and only if for any distribution $\rho$ over $\Theta$ we have that,*
$$KL(\nu(\boldsymbol{x}), p(\boldsymbol{x}|\boldsymbol{\theta}^\star_{ML})) \leq KL(\nu(\boldsymbol{x}), \mathbb{E}_{\rho(\boldsymbol{\theta})}[p(\boldsymbol{x}|\boldsymbol{\theta})]).$$
*And $\rho^\star_{ML}$ can always be characterized as a Dirac-delta distribution center around $\boldsymbol{\theta}^\star_{ML}$, i.e., $\rho^\star_{ML}(\boldsymbol{\theta}) = \delta_{\boldsymbol{\theta}^\star_{ML}}(\boldsymbol{\theta})$.*

*Proof.* We first prove that if the inequality of the lemma holds, then $\rho^\star_{ML}$, is also a minimizer of the cross-entropy loss $CE(\rho)$. Due to the following equality,
$$KL(\nu(\boldsymbol{x}), \mathbb{E}_{\rho(\boldsymbol{\theta})}[p(\boldsymbol{x}|\boldsymbol{\theta})]) = CE(\rho) - H(\nu). \tag{A.8}$$
Any $\rho$ minimizing Equation (A.8) is also a minimum of the cross-entropy loss, $CE(\rho)$, because $H(\nu)$ is constant w.r.t. $\rho$. And, according to Lemma A.6, any density $\rho^\star_{ML}$ minimizing the expected log-loss satisfies that $\mathbb{E}_{\rho^\star_{ML}(\boldsymbol{\theta})}[p(\boldsymbol{x}|\boldsymbol{\theta})] = p(\boldsymbol{x}|\boldsymbol{\theta}^\star_{ML})$.

So, if the inequality of Lemma 2 holds, $\rho^\star_{ML}$ is also a minimum of Equation (A.8) and, as a consequence, $\rho^\star_{ML}$ is also a minimum of the cross-entropy loss, $CE(\rho)$.

We now have to prove that if $\rho^\star_{ML}$ is also a minimizer of the cross-entropy loss $CE(\rho)$, then the inequality of the lemma also holds. Equivalently, we have that if the inequality of the lemma does not hold, then $\rho^\star_{ML}$ can not be a minimizer of Equation (A.8) and, as a consequence, is not a minimizer of $CE(\rho)$.

Finally, according to Lemma A.6 we always have that $\mathbb{E}_{\rho^\star_{ML}(\boldsymbol{\theta})}[p(\boldsymbol{x}|\boldsymbol{\theta})] = p(\boldsymbol{x}|\boldsymbol{\theta}^\star_{ML})$, so $\rho^\star_{ML}$ can always be characterized as a Dirac-delta distribution center around $\boldsymbol{\theta}^\star_{ML}$, i.e., $\rho^\star_{ML}(\boldsymbol{\theta}) = \delta_{\boldsymbol{\theta}^\star_{ML}}(\boldsymbol{\theta})$. $\qquad\square$

## A.5 Proof of Theorem 2

**Theorem 2.** *Any distribution $\rho$ over $\boldsymbol{\Theta}$ satisfies that,*

$$CE(\rho) \leq \mathbb{E}_{\rho(\boldsymbol{\theta})}[L(\boldsymbol{\theta})] - \mathbb{V}(\rho),$$

*where $\mathbb{V}(\rho)$ is a variance term defined as $\mathbb{V}(\rho) = \mathbb{E}_{\nu(\boldsymbol{x})}[\frac{1}{2\max_{\boldsymbol{\theta}} p(\boldsymbol{x}|\boldsymbol{\theta})^2} \mathbb{E}_{\rho(\boldsymbol{\theta})}[(p(\boldsymbol{x}|\boldsymbol{\theta}) - p(\boldsymbol{x}))^2]]$, where $\max_{\boldsymbol{\theta}} p(\boldsymbol{x}|\boldsymbol{\theta})^2$ is a finite scalar value according to Assumption 1.*

*Proof.* Applying Taylor's theorem to $\ln y$ about $\mu$ with a mean-value form of the remainder gives,

$$\ln y = \ln \mu + \frac{1}{\mu}(y - \mu) - \frac{1}{2g(y)^2}(y - \mu)^2,$$

where $g(y)$ is a real value between $y$ and $\mu$. Therefore,

$$\mathbb{E}_{\rho(\boldsymbol{\theta})}[\ln p(\boldsymbol{x}|\boldsymbol{\theta})] = \ln \mathbb{E}_{\rho(\boldsymbol{\theta})}[p(\boldsymbol{x}|\boldsymbol{\theta})] - \mathbb{E}_{\rho(\boldsymbol{\theta})}[\frac{1}{2g(p(\boldsymbol{x}|\boldsymbol{\theta}))^2}(p(\boldsymbol{x}|\boldsymbol{\theta}) - p(\boldsymbol{x}))^2]$$

Rearranging we have

$$-\ln \mathbb{E}_{\rho(\boldsymbol{\theta})}[p(\boldsymbol{x}|\boldsymbol{\theta})] = -\mathbb{E}_{\rho(\boldsymbol{\theta})}[\ln p(\boldsymbol{x}|\boldsymbol{\theta})] - \mathbb{E}_{\rho(\boldsymbol{\theta})}[\frac{1}{2g(p(\boldsymbol{x}|\boldsymbol{\theta}))^2}(p(\boldsymbol{x}|\boldsymbol{\theta}) - p(\boldsymbol{x}))^2]$$

$$\leq -\mathbb{E}_{\rho(\boldsymbol{\theta})}[\ln p(\boldsymbol{x}|\boldsymbol{\theta})] - \frac{1}{2\max_{\boldsymbol{\theta}} p(\boldsymbol{x}|\boldsymbol{\theta})^2} \mathbb{E}_{\rho(\boldsymbol{\theta})}[(p(\boldsymbol{x}|\boldsymbol{\theta}) - p(\boldsymbol{x}))^2],$$

where the inequality is derived from that fact $(p(\boldsymbol{x}|\boldsymbol{\theta}) - p(\boldsymbol{x}))^2$ is always positive and that for all $\boldsymbol{\theta} \in supp(\rho)$ the term $g(p(\boldsymbol{x}|\boldsymbol{\theta}))$, which is a real number between $p(\boldsymbol{x}|\boldsymbol{\theta})$ and $\mathbb{E}_{\rho(\boldsymbol{\theta})}[p(\boldsymbol{x}|\boldsymbol{\theta})]$, is upper bounded by $\max_{\boldsymbol{\theta}} p(\boldsymbol{x}|\boldsymbol{\theta})$. Finally, the result of the theorem is derived by taking expectation wrt $\nu(\boldsymbol{x})$ on both sides of the above inequality. $\square$

## A.6 Proof of Lemma 3

Before proving Lemma 3, we need to introduce the following preliminary result,

**Lemma A.7.** *If a density $\rho'$ over $\boldsymbol{\Theta}$ has null variance, i.e. $\mathbb{V}(\rho') = 0$, where $\mathbb{V}(\rho')$ is defined in Theorem 2, then we have the following equality,*

$$CE(\rho') = \mathbb{E}_{\rho'(\boldsymbol{\theta})}[L(\boldsymbol{\theta})]$$

*Proof.* We first have that if $\mathbb{V}(\rho') = 0$, then all parameters in the support of $\rho'$ induce the same probability distribution over $\boldsymbol{x}$. Because the variance term will be not null as soon as we have two parameters in the support of $\rho'$ which induces two different probability distributions over $\boldsymbol{x}$. We then denote $\boldsymbol{\theta}'$ to a parameter in the support of $\rho'$, i.e., $\boldsymbol{\theta}' \in supp(\rho')$.

In this case, we can deduce that if $\mathbb{V}(\rho') = 0$, then $\mathbb{E}_{\rho'(\boldsymbol{\theta})}[p(\boldsymbol{x}|\boldsymbol{\theta})] = \mathbb{E}_{\delta_{\boldsymbol{\theta}'}(\boldsymbol{\theta})}[p(\boldsymbol{x}|\boldsymbol{\theta})]$, because all the parameters in the support of $\rho'$ induce the same distribution over $\boldsymbol{x}$. So, $\rho'$ can be reparametrized as a Dirac-delta distribution. And, by Equation (A.7), we have that $\mathbb{E}_{\delta_{\boldsymbol{\theta}'}(\boldsymbol{\theta})}[p(\boldsymbol{x}|\boldsymbol{\theta})] = p(\boldsymbol{x}|\boldsymbol{\theta}')$.

Finally, we have that,

$$\mathbb{E}_{\rho'(\boldsymbol{\theta})}[L(\boldsymbol{\theta})] = L(\boldsymbol{\theta}') = \mathbb{E}_{\nu(\boldsymbol{x})}[\ln \frac{1}{p(\boldsymbol{x}|\boldsymbol{\theta}')}] = \mathbb{E}_{\nu(\boldsymbol{x})}[\ln \frac{1}{\mathbb{E}_{\rho'(\boldsymbol{\theta})}[p(\boldsymbol{x}|\boldsymbol{\theta})]}] = CE(\rho'),$$

where the first equality follows because, as we have seen above, $\rho'$ acts as a Dirac-delta distributions (see Equation (A.7)), the second equality follows from the definition of $L(\boldsymbol{\theta})$, the third equality follows again from the property of Dirac-delta distributions, and the last equality follows from the definition of the cross-entropy loss $CE$. $\square$

We now introduce the proof of Lemma 3.

**Lemma 3.** *Let us denote $\rho_{J2}^\star$ and $\rho_{ML}^\star$ a distribution minimizing the second-order Jensen bound of Theorem 2 and $\mathbb{E}_{\rho(\boldsymbol{\theta})}[L(\boldsymbol{\theta})]$, respectively. The following inequality holds*

$$KL(\nu(\boldsymbol{x}), \mathbb{E}_{\rho_{J2}^\star}[p(\boldsymbol{x}|\boldsymbol{\theta})]) \leq KL(\nu(\boldsymbol{x}), \mathbb{E}_{\rho_{ML}^\star}[p(\boldsymbol{x}|\boldsymbol{\theta})]).$$

*Under perfect model specification or in a degenerated model averaging setting (see Lemma 2) both KL terms are equal.*

*Proof.* Let us define $\Delta$ the space of distributions $\rho$ over $\boldsymbol{\Theta}$ whose variance is null, i.e., if $\rho \in \Delta$ then $\mathbb{V}(\rho') = 0$, where $\mathbb{V}(\rho')$ is defined in Theorem 2. Then, we have that the minimum of the second-order Jensen bound for all the distributions $\rho \in \Delta$ can be written as,

$$\min_{\rho \in \Delta} \mathbb{E}_{\rho(\boldsymbol{\theta})}[L(\boldsymbol{\theta})] - \mathbb{V}(\rho) = \min_{\rho \in \Delta} \mathbb{E}_{\rho(\boldsymbol{\theta})}[L(\boldsymbol{\theta})] = \mathbb{E}_{\rho^\star_{ML}(\boldsymbol{\theta})}[L(\boldsymbol{\theta})],$$

where the first inequality follows because if $\rho \in \Delta$ then $\mathbb{V}(\rho) = 0$, and the second equality follows from Lemma A.6. We also have that

$$\mathbb{E}_{\rho^\star_{J^2}(\boldsymbol{\theta})}[L(\boldsymbol{\theta})] - \mathbb{V}(\rho^\star_{J^2}) \le \mathbb{E}_{\rho^\star_{ML}(\boldsymbol{\theta})}[L(\boldsymbol{\theta})] \tag{A.9}$$

because, by definition, the left hand side of the inequality is the minimum of the second-order Jensen bound for all the distributions $\rho(\boldsymbol{\theta})$ over $\boldsymbol{\Theta}$, while the right hand side of the inequality is the minimum of the second-order Jensen bound but only for those distributions $\rho \in \Delta$.

By chaining the above inequality of Equation (A.9) with the second-order Jensen bound inequality of Theorem 2, we have

$$CE(\rho^\star_{J^2}) \le \mathbb{E}_{\rho^\star_{ML}(\boldsymbol{\theta})}[L(\boldsymbol{\theta})] \tag{A.10}$$

Finally, we have that $\mathbb{E}_{\rho^\star_{ML}(\boldsymbol{\theta})}[L(\boldsymbol{\theta})] = CE(\rho^\star_{ML})$ by Lemma A.7 because $\mathbb{V}(\rho^\star_{ML}) = 0$ due to Lemma A.6. So, combining $\mathbb{E}_{\rho^\star_{ML}(\boldsymbol{\theta})}[L(\boldsymbol{\theta})] = CE(\rho^\star_{ML})$ with the inequality of Equation (A.10), we have that,

$$CE(\rho^\star_{J^2}) \le CE(\rho^\star_{ML}) \tag{A.11}$$

which proofs the inequality of the lemma, because $KL(\nu(\boldsymbol{x}), \mathbb{E}_{\rho(\boldsymbol{\theta})}[p(\boldsymbol{x}|\boldsymbol{\theta})]) = CE(\rho) - H(\nu)$.

If we are under the conditions of Lemma 2, we have that for any distribution $\rho$ over $\boldsymbol{\Theta}$, $KL(\nu(\boldsymbol{x}), p(\boldsymbol{x}|\boldsymbol{\theta}^\star_{ML})) \le KL(\nu(\boldsymbol{x}), \mathbb{E}_{\rho(\boldsymbol{\theta})}[p(\boldsymbol{x}|\boldsymbol{\theta})])$. From this condition, we can deduce that for any distribution $\rho$ over $\boldsymbol{\Theta}$,

$$C(\rho^\star_{ML}) \le CE(\rho), \tag{A.12}$$

because $KL(\nu(\boldsymbol{x}), \mathbb{E}_{\rho(\boldsymbol{\theta})}[p(\boldsymbol{x}|\boldsymbol{\theta})]) = CE(\rho) - H(\nu)$ and $KL(\nu(\boldsymbol{x}), p(\boldsymbol{x}|\boldsymbol{\theta}^\star_{ML})) = KL(\nu(\boldsymbol{x}), \mathbb{E}_{\rho^\star_{ML}}[p(\boldsymbol{x}|\boldsymbol{\theta})]) = C(\rho^\star_{ML}) - H(\nu)$. Combining Equations (A.10) and (A.12), we have that $CE(\rho^\star_{J^2}) = C(\rho^\star_{ML})$ under the conditions of Lemma 2, proving that the inequality of the lemma becomes an equality. $\square$

Characterizing under which conditions the inequality of Lemma 3 becomes strict is an open problem and an interesting subject of future research.

## A.7 Proof of Theorem 3

Before proving Theorem 3, we need to introduce the following result,

**Lemma A.8.** *For any prior distribution $\pi$ over $\boldsymbol{\Theta}$ and for any distribution $\rho$ over $\boldsymbol{\Theta}$, the second-order Jensen bound of Theorem 2 bound can be expressed as follows,*

$$\mathbb{E}_{\rho(\boldsymbol{\theta})}[L(\boldsymbol{\theta})] - \mathbb{V}(\rho) = \mathbb{E}_{\rho(\boldsymbol{\theta},\boldsymbol{\theta}')}[L(\boldsymbol{\theta}, \boldsymbol{\theta}')],$$

*where $\boldsymbol{\theta}, \boldsymbol{\theta}' \in \boldsymbol{\Theta}$, $\rho(\boldsymbol{\theta}, \boldsymbol{\theta}') = \rho(\boldsymbol{\theta})\rho(\boldsymbol{\theta}')$, and $L(\boldsymbol{\theta}, \boldsymbol{\theta}')$ is defined as*

$$L(\boldsymbol{\theta}, \boldsymbol{\theta}') = \mathbb{E}_{\nu(\boldsymbol{x})}[\ln \frac{1}{p(\boldsymbol{x}|\boldsymbol{\theta})} - \frac{1}{2 \max_{\boldsymbol{\theta}} p(\boldsymbol{x}|\boldsymbol{\theta})^2} \left( p(\boldsymbol{x}|\boldsymbol{\theta})^2 - p(\boldsymbol{x}|\boldsymbol{\theta})p(\boldsymbol{x}|\boldsymbol{\theta}') \right)],$$

*Proof.* The proof is straightforward by applying first this equality,

$$\mathbb{E}_{\rho(\boldsymbol{\theta})}[(p(\boldsymbol{x}|\boldsymbol{\theta}) - p(\boldsymbol{x}))^2] = \mathbb{E}_{\rho(\boldsymbol{\theta})}[p(\boldsymbol{x}|\boldsymbol{\theta})^2] - \mathbb{E}_{\rho(\boldsymbol{\theta})}[p(\boldsymbol{x}|\boldsymbol{\theta})]^2,$$

and, after that, the following equality,

$$\mathbb{E}_{\rho(\boldsymbol{\theta},\boldsymbol{\theta}')}[p(\boldsymbol{x}|\boldsymbol{\theta})p(\boldsymbol{x}|\boldsymbol{\theta}')] = \mathbb{E}_{\rho(\boldsymbol{\theta})}[p(\boldsymbol{x}|\boldsymbol{\theta})\mathbb{E}_{\rho(\boldsymbol{\theta}')}[p(\boldsymbol{x}|\boldsymbol{\theta}')]] = \mathbb{E}_{\rho(\boldsymbol{\theta})}[p(\boldsymbol{x}|\boldsymbol{\theta})]^2$$

$\square$

We now proceed to the proof of Theorem 3.

**Theorem 3.** *For any prior distribution $\pi$ over $\Theta$ independent of $D$ and for any $\xi \in (0,1)$ and $c > 0$, with probability at least $1 - \xi$ over draws of training data $D \sim \nu^n(\boldsymbol{x})$, for all distribution $\rho$ over $\Theta$, simultaneously,*

$$CE(\rho) \leq \mathbb{E}_{\rho(\boldsymbol{\theta})}[L(\boldsymbol{\theta})] - \mathbb{V}(\rho) \leq \mathbb{E}_{\rho(\boldsymbol{\theta})}[\hat{L}(\boldsymbol{\theta}, D)] - \hat{\mathbb{V}}(\rho, D) + \frac{KL(\rho, \pi) + \frac{1}{2}\ln\frac{1}{\xi} + \frac{1}{2}\psi'_{\pi,\nu}(c,n)}{cn},$$

*where $\psi'_{\pi,\nu}(c,n)$ is the same term as in Theorem 1 adapted to this setting and $\hat{\mathbb{V}}(\rho, D)$ is the empirical version of $\mathbb{V}(\rho)$.*

*Proof.* By Lemma A.8, we can express the problem using a *tandem log-loss* $L(\boldsymbol{\theta}, \boldsymbol{\theta}')$. Then, we apply [18, Theorem 3] to this loss using as prior $\pi(\boldsymbol{\theta}, \boldsymbol{\theta}) = \pi(\boldsymbol{\theta})\pi(\boldsymbol{\theta}')$ and have

$$\mathbb{E}_{\rho(\boldsymbol{\theta}, \boldsymbol{\theta}')}[L(\boldsymbol{\theta}, \boldsymbol{\theta}')] \leq \mathbb{E}_{\rho(\boldsymbol{\theta}, \boldsymbol{\theta}')}[\hat{L}(\boldsymbol{\theta}, \boldsymbol{\theta}', D)] + \frac{1}{\lambda}(KL(\rho(\boldsymbol{\theta}, \boldsymbol{\theta}'), \pi(\boldsymbol{\theta}, \boldsymbol{\theta}')) + \ln\frac{1}{\xi} + \psi'_{\pi,\nu}(\lambda, n)),$$

where $\psi'_{\pi,\nu}(\lambda, n)$ is defined as

$$\psi'_{\pi,\nu}(\lambda, n) = \ln \mathbb{E}_{\pi(\boldsymbol{\theta}, \boldsymbol{\theta}')}\mathbb{E}_{D \sim \nu^n(\boldsymbol{x})}[e^{\lambda(L(\boldsymbol{\theta}, \boldsymbol{\theta}') - \hat{L}(\boldsymbol{\theta}, \boldsymbol{\theta}', D))}].$$

The PAC-Bayes bound of the theorem follows by rewriting $\mathbb{E}_{\rho(\boldsymbol{\theta}, \boldsymbol{\theta}')}[L(\boldsymbol{\theta}, \boldsymbol{\theta}')] = \mathbb{E}_{\rho(\boldsymbol{\theta})}[L(\boldsymbol{\theta})] - \mathbb{V}(\rho)$ and $\mathbb{E}_{\rho(\boldsymbol{\theta}, \boldsymbol{\theta}')}[\hat{L}(\boldsymbol{\theta}, \boldsymbol{\theta}', D)] = \mathbb{E}_{\rho(\boldsymbol{\theta})}[\hat{L}(\boldsymbol{\theta}, D)] - \hat{\mathbb{V}}(\rho, D)$, and noting that $KL(\rho(\boldsymbol{\theta}, \boldsymbol{\theta}'), \pi(\boldsymbol{\theta}, \boldsymbol{\theta}')) = 2KL(\rho(\boldsymbol{\theta}), \pi(\boldsymbol{\theta}))$. Finally, we reparametrized $\lambda$ as $\lambda = 2cn$. $\square$

## B First-order vs second-order Jensen bounds

Figure 2 illustrates the behavior of first-order and second-order Jensen bounds under perfect model specification and model-misspecification. In this case, the model space is composed of two Binomial models. The $\rho$ distribution can be defined with a single parameter between 0 and 1. Extremes ($\rho = 0$ or $\rho = 1$) values choose a single model. Non-extremes values induce an averaging of the models. Left figure shows the situation of model misspecification because there exists an optimal $\rho$ distribution minimizing the cross-entropy function (i.e., achieving optimal prediction performance), but the expected log-loss is minimized with a Dirac-delta distribution (i.e., $\rho = 1$), choosing the best single model. While the minimum of the second-order Jensen bound achieves a better result. Right figure shows the case of perfect model specification. In this case, the data generating distribution corresponds to one of the models, $\rho = 1$ is the distribution minimizing the cross-entropy loss (i.e., achieving optimal prediction performance). In this case, both the expected log-loss and the second-order Jensen bound are minimized with a Dirac-delta distribution (i.e., $\rho = 1$), choosing the best single model and achieving optimal results.

## C Learning algorithms

### C.1 Tighter Jensen Bounds

The next result shows how we can define a tighter second-order Jensen bound using the Jensen inequality presented in [35].

**Theorem C.5.** *Any distribution $\rho$ over $\Theta$ satisfies the following inequality,*

$$CE(\rho) \leq \mathbb{E}_{\rho(\boldsymbol{\theta})}[L(\boldsymbol{\theta})] - \mathbb{V}_T(\rho),$$

*where $\mathbb{V}_T(\rho)$ is the normalized variance of $p(\boldsymbol{x}|\boldsymbol{\theta})$ wrt $\rho(\boldsymbol{\theta})$,*

$$\mathbb{V}_T(\rho) = \mathbb{E}_{\nu(\boldsymbol{x})}[h(m_{\boldsymbol{x}}, \mu_{\boldsymbol{x}})\mathbb{E}_{\rho(\boldsymbol{\theta})}[(p(\boldsymbol{x}|\boldsymbol{\theta}) - p(\boldsymbol{x}))^2]].$$

*and $\mu_{\boldsymbol{x}} = \mathbb{E}_{\rho(\boldsymbol{\theta})}[p(\boldsymbol{x}|\boldsymbol{\theta})]$, $m_{\boldsymbol{x}} = \max_{\boldsymbol{\theta}} p(\boldsymbol{x}|\boldsymbol{\theta})$ and $h(m, \mu) = \frac{\ln\mu - \ln m}{(m-\mu)^2} + \frac{1}{\mu(m-\mu)}$*

*Proof sketch.* Apply [35]'s result to the random variable $p(\boldsymbol{x}|\boldsymbol{\theta})$, following the same strategy used in the proof of Theorem 2. $\square$

As shown in [35], we have that $\forall \boldsymbol{x} \in \mathcal{X} \quad h(m_{\boldsymbol{x}}, \mu_{\boldsymbol{x}}) \geq \frac{1}{2 \max_{\boldsymbol{\theta}} p(\boldsymbol{x}|\boldsymbol{\theta})^2}$. In consequence, the above bound is tighter because $\mathbb{V}_T(\rho) \geq \mathbb{V}(\rho)$ (see Theorem 2). Similarly, we can show that the same inequality applies for the empirical versions of both variance terms, i.e., $\hat{\mathbb{V}}_T(\rho, D) \geq \hat{\mathbb{V}}(\rho, D)$.

The issue with the introduction of the $\mathbb{V}_T(\rho)$ term is that we can not derive the corresponding second-order PAC-Bayes bound using the same approach of Theorem 3. This is, therefore, an open issue.

## C.2 PAC$^2$-Variational Learning

The PAC$^2$-Variational learning algorithm is based on the optimization of Equation (4). Similarly, the PAC$_T^2$-Variational algorithm is based on the minimization of the same equation but replacing the $\hat{\mathbb{V}}(\rho, D)$ term with the $\hat{\mathbb{V}}_T(\rho, D)$ term (see Equation (5)). We can rewrite Equation (4), by employing the expression provided in Lemma A.8, to have an amenable and numerically stable version. We do it by multiplying and dividing the variance term by $2 \max_{\boldsymbol{\theta}} p(\boldsymbol{x}|\boldsymbol{\theta})^2$ and, also, by multiplying the whole expression by $n$. So, Equation (4) can be expressed as follows,

$$\mathbb{E}_{\rho(\boldsymbol{\theta}, \boldsymbol{\theta}'|\boldsymbol{\lambda})}[\sum_{i=1}^{n} - \ln p(\boldsymbol{x}_i|\boldsymbol{\theta}) - h(\alpha_{\boldsymbol{x}_i}) \hat{\mathbb{V}}(\boldsymbol{x}_i, \boldsymbol{\theta}, \boldsymbol{\theta}')] + KL(\rho, \pi) \tag{C.13}$$

where $\rho(\boldsymbol{\theta}, \boldsymbol{\theta}'|\boldsymbol{\lambda}) = \rho(\boldsymbol{\theta}|\boldsymbol{\lambda}) \rho(\boldsymbol{\theta}'|\boldsymbol{\lambda})$ and

$$m_{\boldsymbol{x}_i} = \max_{\boldsymbol{\theta}} \ln p(\boldsymbol{x}_i|\boldsymbol{\theta})$$
$$\mathbb{V}(\boldsymbol{x}_i, \boldsymbol{\theta}, \boldsymbol{\theta}') = \exp(2 \ln p(\boldsymbol{x}_i|\boldsymbol{\theta}) - 2m_{\boldsymbol{x}_i}) - \exp(\ln p(\boldsymbol{x}_i|\boldsymbol{\theta}) + \ln p(\boldsymbol{x}_i|\boldsymbol{\theta}') - 2m_{\boldsymbol{x}_i}).$$

For the PAC$^2$-Variational learning algorithm we have that $h(\alpha_{\boldsymbol{x}}) = 1$, but for the PAC$_T^2$-Variational learning algorithm, we have that

$$h(\alpha_{\boldsymbol{x}}) = \frac{\alpha_{\boldsymbol{x}}}{(1 - \exp(\alpha_{\boldsymbol{x}}))^2} + \frac{1}{\exp(\alpha_{\boldsymbol{x}})(1 - \exp(\alpha_{\boldsymbol{x}}))}$$
$$\alpha_{\boldsymbol{x}_i} = \ln(\exp(\ln p(\boldsymbol{x}_i|\boldsymbol{\theta}) - m_{\boldsymbol{x}_i}) + \exp(\ln p(\boldsymbol{x}_i|\boldsymbol{\theta}') - m_{\boldsymbol{x}_i})) - \ln 2.$$

For supervised classification problems, we fix $m_{\boldsymbol{x}_i} = 0$, assuming it is possible to make always a perfect classification. For regression tasks, we sample two parameters[7] $\boldsymbol{\theta}, \boldsymbol{\theta}' \sim \rho(\boldsymbol{\theta}|\boldsymbol{\lambda})$ and take $m_{\boldsymbol{x}_i} = \max(\ln p(\boldsymbol{x}_i|\boldsymbol{\theta}), \ln p(\boldsymbol{x}_i|\boldsymbol{\theta}')) + \epsilon$, with $\epsilon = 0.1$ to avoid numerically stability problems when computing $h(\alpha_{\boldsymbol{x}})$. Even though, better strategies can be defined to compute $m_{\boldsymbol{x}_i}$.

We can minimize Equation (C.13) using any gradient-based optimizing algorithm. Unbiased estimates of the gradient of Equation (C.13) can be computed using appropriate Monte-Carlo gradient estimation methods [43]. We apply *stop-gradient* operation over $m_{\boldsymbol{x}_i}$ and $h(\alpha_{\boldsymbol{x}_i})$ to avoid problems deriving a *max* or a *log-sum-exp* operation.

Figures C.5 and C.6 compare the Bayesian/Variational posterior and the Bayesian/Variational posterior predictive distribution and its PAC$^2$-Variational counter-parts for a simple misspecified linear model and for a neural network based model using sinusoidal data, respectively. Figure C.8 also illustrates the behavior of the PAC$^2$-Variational learning methods under perfect model specification. In this case, we can observe how PAC$^2$-Variational methods recover the Bayesian posterior and agree with standard variational methods under perfect model specification in both models (i.e. the posterior predictive of the PAC$^2$-Variational algorithm matches the posterior predictive distribution of standard Bayesian/Variational methods).

## C.3 PAC$^2$-Ensemble Learning

We first start providing a proof of Theorem 4,

Figure C.5: **Linear Model**: The data generating model, $\nu(y|x)$, is $y \sim \mathcal{N}(\mu = 1 + x, \sigma^2 = 5)$. The probabilistic model, $p(y|x, \boldsymbol{\theta})$ is $y \sim \mathcal{N}(\mu = \theta_0 + \theta_1 x, \sigma^2 = 1)$. So, the probabilistic model is miss-specified, but note how model misspecification mainly affects to the parameter $\theta_0$. The prior $\pi(\boldsymbol{\theta})$ is a standard Normal distribution. We learn from 100 samples. The Bayesian posterior $p(\boldsymbol{\theta}|D)$ is a bidimensional Normal distribution and can be exactly computed. The $\text{PAC}_T^2$-Variational posterior is computed using a bidimensional Normal distribution approximation family and the Adam optimizer. The uncertainty in the predictions is computed by random sampling first from $\boldsymbol{\theta} \sim \rho(\boldsymbol{\theta})$ and then from $y \sim p(y|x, \boldsymbol{\theta})$ for 100 times. We plot the predictive mean plus/minus two standard deviations. We distinguish between *epistemic uncertainty* which comes from the uncertainty in $\rho(\boldsymbol{\theta})$ and *aleatory uncertainty* which comes from the uncertainty in $p(y|x, \boldsymbol{\theta})$. The test log-likelihood of the posterior predictive distribution is -14.25, -13.09, -10.56 and -7.89 for the MAP, the Bayesian, the $\text{PAC}^2$-Variational and the $\text{PAC}_T^2$-Variational posterior predictive distributions, respectively. The test log-likelihood is computed from an independent test set of 10000 samples.

**Theorem 4.** *For any prior distribution $\pi_F$ over $\boldsymbol{\Theta}_F$ and independent of $D$ and for any $\xi \in (0, 1)$ and $c > 0$, with probability at least $1 - \xi$ over draws of training data $D \sim \nu^n(\boldsymbol{x})$, for all densities $\rho_E$ with $supp(\rho_E) \subseteq \boldsymbol{\Theta}_F$, simultaneously,*

$$CE(\rho_E) \leq \mathbb{E}_{\rho_E(\boldsymbol{\theta})}[\hat{L}(\boldsymbol{\theta}, D)] - \hat{\mathbb{V}}(\rho_E, D) + \frac{KL(\rho_E, \pi_F) + \frac{1}{2}\ln\frac{1}{\xi} + \frac{1}{2}\psi'_{\pi_F, \nu}(c, n)}{cn},$$

*where $\psi'_{\pi_F, \nu}(c, n)$ is the same term as in Theorem 3, and $\hat{\mathbb{V}}(\rho_E, D)$ is the empirical variance,*

$$\hat{\mathbb{V}}(\rho_E, D) = \frac{1}{nE}\sum_{i=1}^{n}\sum_{j=1}^{E}\frac{(p(\boldsymbol{x}_i|\boldsymbol{\theta}_j) - p_E(\boldsymbol{x}_i))^2}{2\max_{\boldsymbol{\theta}} p(\boldsymbol{x}_i|\boldsymbol{\theta})^2}.$$

*Proof.* The result follows from Theorem 3 when considering a density $\rho_E$ and a prior $\pi_F$. The KL distance between $\rho_E$ and $\pi_F$ is well defined because,

$$KL(\rho_E, \pi_F) = \sum_{j=1}^{E}\frac{1}{E}\ln\frac{\frac{1}{E}\delta_{\boldsymbol{\theta}_j}(\boldsymbol{\theta}_j)}{w_{\boldsymbol{\theta}_j}\delta_{\boldsymbol{\theta}_j}(\boldsymbol{\theta}_j)} = \frac{1}{E}\sum_{j=1}^{E}\ln\frac{\frac{1}{E}}{w_{\boldsymbol{\theta}_j}}.$$

$\square$

Theorem 4 shows how to design a learning algorithm for building ensembles by minimizing the PAC-Bayes upper bound. The algorithm we propose is obtained by fixing $c = 1$ and discarding

Figure C.6: **Neural Network Model:** The data generating model, $\nu(y|x)$, is a sinusoidal function plus Gaussian noise, $y = s(x) + \epsilon$, $\epsilon \sim \mathcal{N}(0, \sigma^2 = 10)$. The probabilistic model $p(y|x, \boldsymbol{\theta})$ is $y = f_{\boldsymbol{\theta}}(x) + \epsilon$, $\epsilon \sim \mathcal{N}(0, \sigma^2 = 1)$, where $f$ is a MLP parametrized by $\boldsymbol{\theta}$, with one hidden layer with 20 units able to approximate $s(x)$. The prior $\pi(\boldsymbol{\theta})$ is a standard Normal distribution. 10000 training samples. The Variational and the PAC$^2$-Variational approximation family $Q$ is a fully factorized mean field Normal distribution and it is optimized with the Adam optimizer. The test log-likelihood of the posterior predictive distribution is -50.44, -50.15, -36.60 and -25.23 for the MAP, the Variational, the PAC$^2$-Variational and the PAC$^2_T$-Variational models, respectively. While for the PAC$^2$-Ensemble and the PAC$^2_T$-Ensemble models is -38.95 and -15.91, respectively. Uncertainty estimations are computed as in Figure C.5.

constant terms wrt $\rho_E$. We call it the *PAC$^2$-Ensemble learning algorithm.* So, learning the ensemble reduces to find the parameters $\{\boldsymbol{\theta}_j\}_{1 \leq j \leq E}$ that minimize the following objective function,

$$\arg \min_{\{\boldsymbol{\theta}_1, \dots, \boldsymbol{\theta}_E\}} \frac{1}{E} \sum_{j=1}^{E} \hat{L}(\boldsymbol{\theta}_j, D) - \hat{\mathbb{V}}(\rho_E, D) - \frac{1}{E} \sum_{j=1}^{E} \frac{\ln \pi_F(\boldsymbol{\theta}_j)}{n} \qquad \text{(C.14)}$$

Similarly to PAC$^2$-Variational algorithms, we can also employ the tighter second-order Jensen bound mentioned in Section 6.1 to derive a new objective function by replacing $\hat{\mathbb{V}}(\rho_E, D)$ with $\hat{\mathbb{V}}_T(\rho_E, D)$ (see Equation (5)). We call this algorithm *PAC$^2_T$-Ensemble learning.*

We have that Equation (C.14) is a non-continuous and non-differentiable function due to the $\ln \pi_F(\boldsymbol{\theta})$ term. But in a computer, any implemented statistical distribution $\pi(\boldsymbol{\theta})$ (e.g. a Normal distribution) can be seen as an approximation for its finite-precision counter-part, $\pi_F(\boldsymbol{\theta})$. So, we can employ this continuous and differentiable approximation $\ln \pi(\boldsymbol{\theta})$ as a proxy to perform gradient-based optimization involving the term $\ln \pi_F(\boldsymbol{\theta})$. Note that, at each step, our optimization algorithm will end up in a parameter in $\boldsymbol{\Theta}_F$ because they are the only ones which can be represented in the computer.

We show how to express Equation (C.14) in a numerically stable way, using the same strategy employed in Appendix C.2,

$$\sum_{i=1}^{n} \sum_{j=1}^{E} -\ln p(\boldsymbol{x}_i|\boldsymbol{\theta}_j) - h(\alpha_{\boldsymbol{x}_i}) \exp(2 \ln p(\boldsymbol{x}_i|\boldsymbol{\theta}_j) - 2m_{\boldsymbol{x}}) - \ln \pi(\boldsymbol{\theta}_j)$$

$$+ \sum_{k=1}^{E} h(\alpha_{\boldsymbol{x}_i}) \exp(\ln p(\boldsymbol{x}_i|\boldsymbol{\theta}_j) + \ln p(\boldsymbol{x}_i|\boldsymbol{\theta}_k) - 2m_{\boldsymbol{x}})$$

where $m_{\boldsymbol{x}_i} = \max_{\boldsymbol{\theta}} \ln p(\boldsymbol{x}_i|\boldsymbol{\theta})$. For the PAC$^2$-Ensemble learning algorithm we have that $h(\alpha_{\boldsymbol{x}}) = 1$, but for the PAC$_T^2$-Ensemble learning algorithm, we have that

$$\alpha_{\boldsymbol{x}_i} = \ln \sum_{j=1}^{E} \exp(\ln p(\boldsymbol{x}_i|\boldsymbol{\theta}_j) - m_{\boldsymbol{x}_i}) - \ln E$$

$$h(\alpha_{\boldsymbol{x}}) = \frac{\alpha_{\boldsymbol{x}}}{(1 - \exp(\alpha_{\boldsymbol{x}}))^2} + \frac{1}{\exp(\alpha_{\boldsymbol{x}})(1 - \exp(\alpha_{\boldsymbol{x}}))}.$$

Again, for supervised classification problems, we fix $m_{\boldsymbol{x}_i} = 0$, assuming it is possible to make always a perfect classification. For the rest of the cases, we take $m_{\boldsymbol{x}_i} = \max_j \ln p(\boldsymbol{x}_i|\boldsymbol{\theta}_j)$. We also apply *stop-gradient* operation over $m_{\boldsymbol{x}}$ and $h(\alpha_{\boldsymbol{x}_i})$ to avoid problems deriving a *max* or a *log-sum-exp* operation.

As commented at the end of Section 6.3, we can similarly derive an ensemble algorithm from the PAC-Bayes bound of Theorem 1 using $\rho_E$ and $\pi_E$ densities. In this case, the $\hat{\mathbb{V}}(\rho_E, D)$ term would disappear from the PAC-Bayes bound, and the objective function would be,

$$\arg \min_{\{\boldsymbol{\theta}_1, \dots, \boldsymbol{\theta}_E\}} \frac{1}{E} \sum_{j=1}^{E} \hat{L}(\boldsymbol{\theta}_j, D) - \frac{\ln \pi_F(\boldsymbol{\theta}_j)}{n}. \tag{C.15}$$

The following result shows how a collapsed ensemble with all models equal to the MAP model, which is equivalent to a single ensemble model, is a minimum of this objective function,

**Lemma C.9.** *Let us define, $\boldsymbol{\theta}_{MAP}$ as ,*

$$\boldsymbol{\theta}_{MAP} = \arg \min_{\theta} \hat{L}(\boldsymbol{\theta}, D) - \frac{\ln \pi_F(\boldsymbol{\theta})}{n}.$$

*The vector of $E$ replications of the $\boldsymbol{\theta}_{MAP}$ parameter is a minimum of Equation* (C.15).

*Proof.* Note that Equation (C.15) is the sum of $E$ independent functions, $\hat{L}(\boldsymbol{\theta}_j, D) - \frac{\ln \pi_F(\boldsymbol{\theta}_j)}{n}$. So, the minimum over a single $\boldsymbol{\theta}_j$ equals, by definition, $\boldsymbol{\theta}_{MAP}$. In consequence, the vector of $E$ replications of the $\boldsymbol{\theta}_{MAP}$ parameter is a minimum of Equation (C.15). $\qquad\square$

Figures C.6 illustrates this novel ensemble algorithm on a sinusoidal data sample. And Figure C.7 illustrates the performance of the Variational, PAC$^2$-Variational and PAC$^2$-Ensemble approaches over multimodal data. This last figure shows how variational approaches based on a mean-field Gaussian approximation family are not able to properly capture the multimodality nature of the data. However, ensemble approaches define a more flexible approximate family (i.e., a mixture of Dirac-delta distributions) for the posterior distribution and can perfectly capture this multimodality. Again, we also see like the approach based on tighter second-order Jensen bounds performs much better.

## C.4 Learning under perfect model specification

Lemma 3 (and the subsequent discussion) shows how the optimization of second-order oracle bounds provides the same result than the optimization of first-order oracle bounds under perfect model specification. In Appendix B we illustrate this situation for a toy model.

In this section, we show which is the behavior of the proposed learning algorithms based on the minimization of second-order PAC-Bayes bounds when the model family is not specified. For this reason, we consider the same kind of artificial data that we used to illustrate the behavior of these algorithms in Figure C.5 and Figure 3.

The result of this evaluation is provide in Figures C.8 and C.9. In this case, we can observe that the posterior predictive of the PAC$^2$-Variational and PAC$^2$-Ensembles algorithms closely matches the posterior predictive distribution of standard Bayesian/Variational methods.

Figure C.7: **Multimodal data:** Same settings than in previous Figures C.6. But now the data generative function is a mixture of two sinusoidal functions (i.e. multimodal data) with $\sigma^2 = 1$. The test log-likelihood of the posteriors predictive distribution is -18.79, -13.48, -8.91, -13.34 and -4.00 for the Variational, PAC$^2$-Variational, PAC$^2_T$-Variational, PAC$^2$-Ensemble and PAC$^2_T$-Ensemble models, respectively.

Figure C.8: **Perfect Model Specification I**: Same settings than in Figure C.5, but in this case we are under perfect model specification i.e., $\nu(y|x) = \mathcal{N}(\mu = 1 + x, \sigma^2 = 1)$. The test log-likelihood of the posterior predictive distribution is -1.43, -1.41, -1.42 and -1.41 for the MAP, the Variational, the PAC$^2$-Variational and the PAC$^2_T$ -Variational posterior predictive distributions.

Figure C.9: **Perfect Model Specification II:** Same settings than in Figure C.6, but in this case we are under perfect model specification i.e., the data generating model, $\nu(y|x)$, is a sinusoidal function plus Gaussian noise, $y = s(x) + \epsilon$, $\epsilon \sim \mathcal{N}(0, \sigma^2 = 1)$. The test log-likelihood of the posterior predictive distribution is -1.41, -1.42, -1.45 and -1.44 for the MAP, the Variational, the PAC$^2$-Variational and the PAC$^2_T$-Variational models, respectively. While for the PAC$^2$-Ensemble and the PAC$^2_T$-Ensemble models is -2.51 and -2.56, respectively.

## C.5  Setting the constant $c$

The presence of an arbitrary constant is a commonplace in PAC-Bayes bounds. According to PAC-Bayesian principles, this parameter can also be minimized. There are some specific approaches for doing that [13] as this bound does not simultaneously hold for all $c > 0$ values. In this work, we set $c = 1$ to establish the previously mentioned link between PAC-Bayesian theory and Bayesian statistics [18]. By optimizing this constant $c$, we may get some further increase in performance. However, it would require to bound the $\psi'_{\pi,\nu}(c, n)$ term, because, in this case, this term would be involved in the optimization. We could apply the approaches presented in [1, 18] to provide computable upper bounds over $\psi'_{\pi,\nu}(c, n)$, but it would require strong assumptions and would only apply to a restricted family of models.

Another alternative would be to employ an independent validation data set. In this case, we could perform a grid search and choose the $c$ value which leads to the model with better performance.

## D  Details of the empirical evaluation

For the artificial data sets, we employed the following experimental settings: a multilayer perceptron (MLP) with 20 hidden units and a hyperbolic tangent activation function, the Adam optimizer is used with learning rate 0.01, full-batch training and number of epochs equal to 5000. We also use 100 Monte-Carlo samples to compute the posterior predictive distribution for variational methods.

For the experiments with real data sets, we employ a MLP with 20 hidden units and a relu activation function, the Adam optimizer is used with learning rate 0.001, mini-batches with 100 samples and 100 epochs. We use 20 Monte-Carlo samples to compute the posterior predictive distribution for variational methods. We employ default train and test datasets for the CIFAR-10 and Fashion-MNIST data set.

The images of the CIFAR-10 are transformed to grayscale using *Tensorflow* method, *tf.image.rgb_to_grayscale*. Fashion-MNIST's and CIFAR-10's pixels values are normalized to the range 0-1. For this reason, for the self-supervised task with a Normal data model, we set the scale of the Normal distribution to 1/255.

Table D.1 and Table D.2 show the numerical values of the predictive log-likelihood computed on the independent test sets for the variational and ensemble learning algorithms, respectively. Figure 4 was derived from these values.

| Data Set | Task | MAP | Variational | PAC²-Variational | PAC$_t^2$-Variational |
|---|---|---|---|---|---|
| FASHION-MNIST | Supervised | -4237.11 | -3598.26 | -3610.37 | -3558.94 |
| | Self-Sup. Normal | -4.18866854e+09 | -4.14232586e+09 | -4.09604237e+09 | -3.96681664e+09 |
| | Self-Sup. Binomial | -1072927.05 | -1032353.37 | -1007490.71 | -999141.01 |
| CIFAR10 | Supervised | -18346.32 | -19309.88 | -18685.65 | -19156.56 |
| | Self-Sup. Normal | -6.45892346e+09 | -6.32867424e+09 | -6.07852026e+09 | -5.87570592e+09 |
| | Self-Sup. Binomial | -3060388.53 | -2965045.22 | -2886265.31 | -2789976.94 |

Table D.1: Predictive Log-likelihood values on the test set, $\sum_{i=1}^{T} \ln \mathbb{E}_\rho[p(\boldsymbol{x}_i|\boldsymbol{\theta})]$, for different variational learning algorithms.

| Data Set | Algorithm | Task | 1 model/MAP | 2 models | 3 models |
|---|---|---|---|---|---|
| FASHION-MNIST | PAC²-Ensemble | Supervised | -4237.11 | -4143.05 | -4074.1 |
| | | Self-Sup. Normal | -4.18866854e+09 | -4.18866861e+09 | -3.77023802e+09 |
| | | Self-Sup. Binomial | -1072927.05 | -1072292.32 | -973392.38 |
| | PAC$_T^2$-Ensemble | Supervised | -4237.11 | -3708.31 | -3478.14 |
| | | Self-Sup. Normal | -4.18866854e+09 | -3.66828374e+09 | -3.35477882e+09 |
| | | Self-Sup. Binomial | -1072927.05 | -971425.86 | -909174.82 |
| CIFAR10 | PAC²-Ensemble | Supervised | -18346.32 | -17984.42 | -18065.9 |
| | | Self-Sup. Normal | -6.45892346e+09 | -6.45838707e+09 | -5.79018387e+09 |
| | | Self-Sup. Binomial | -3060388.53 | -3060205.56 | -2764825.16 |
| | PAC$_T^2$-Ensemble | Supervised | -18346.32 | -17616.09 | -17587.97 |
| | | Self-Sup. Normal | -6.45892346e+09 | -5.27628016e+09 | -4.87947741e+09 |
| | | Self-Sup. Binomial | -3060388.53 | -2685734.22 | -2528124.42 |

Table D.2: Predictive Log-likelihood values on the test set, $\sum_{i=1}^{T} \ln \frac{1}{E} \sum_{j=1}^{E} p(\boldsymbol{x}_i|\boldsymbol{\theta}_j)$, for different ensemble learning algorithms with a different number of models.