[Reviews · NeurIPS 2020]

Review 1

Summary and Contributions: POST REBUTTAL: I was happy to see that all important issues I raised were addressed to my satisfaction by the rebuttal. Having reviewed an earlier version of this paper for ICML, I know that the authors will keep the promises made in their rebuttal. One minor issue that the rebuttal addresses incorrectly are the assumptions on loss functions (as they relate to PAC-Bayesian bounds). The claim in the rebuttal that the results hold for 'general unbounded losses' is incorrect. In particular, the probability's exponentially fast convergence towards zero is *not* guaranteed for general unbounded losses, and this is actually made rather clear in the references you provide [17,45]. These references use loss functions that satisfy the typical requirements needed for concentration inequalities/PAC-Bayes bounds (e.g., [17] assumes that the losses are sub-gamma and [45] assumes that they are bounded or sub-gamma). It really irked me that you made this false claim in the rebuttal, but luckily the paper does not make this claim. Please make sure this stays the same for the final version of the paper. In summary, the strongest contribution of this paper is to (i) explore a well-known issue (the poor performance of Bayes posteriors under misspecification) with a new tool [the 2nd order bound] and (ii) to direcly weaponize this tool to suggest a new inference method. Since I think these two parts together make for a great contribution, I would like to see this paper accepted. Given that the authors have promised to fix some the issues I raised in my review, I will raise my score to a 7. --- The paper derives a PAC-Bayesian bound which holds under the typical Bernstein or Hoeffding assumptions. The main innovation consists in the choice of the true (i.e. out-of-sample) risk. Specifically, by drawing the connection to traditional Bayes posteriors (which in a PAC-Bayesian context can be understood as setting the out-of-sample risk to be the expected negative log likelihood), the paper advocates for instead choosing the negative log likelihood of the predictive distribution (see eq. (3)). The resulting bound is tighter than its standard counterpart by an additive factor that directly corresponds to the second-order term of the employed second-order Jensen's IE. The authors advocate the resulting bound as being particularly useful under the case of model misspecification, where the difference between optimizing the scoring function vs optimizing the predictive distribution is most pronounced. Lastly, the paper advertises the practical usefulness of this approach by proposing a variational inference algorithm, a special case of which is an ensemble method (where the prior consists of a sum of dirac deltas). While I enjoyed the rest of the paper, the point of this discretized ensemble method is unclear to me: Variational methods without artificially discretized priors are theoretically more interesting and not less practical by virtue of the advent of large-scale stochastic VI. Further, any variational posterior is already an ensemble so that the distinction between the two is strange.

Strengths: * To the best of my knowledge, applying the proposed second-order IE for the purposes of a PAC-Bayesian analysis is novel. As it yields a strictly smaller generalization bound than the vanilla version of the bound (under the necessary assumptions that render the second order expansion valid), this is both a practical and elegant result. *As the authors point out themselves, misspecification is a pervasive problem in statistical analysis. Tackling the issue by changing the criterion one seeks to optimize (the predictive distribution/CE instead of the usual risk measure associated with a PAC-Bayesian interpretation of Bayesian inference) is intuitive, practical, and useful.

Weaknesses: (1) [I don't perceive the following as weaknesses, but it is a limitation (which I wish were more clearly communicated in the manuscript)] The analysis applies only to losses/likelihoods (and priors) under which the usual Bernstein/Hoeffding type assumptions hold. This essentially confines the applicability to classification problems (and a fairly limited class of sub-gaussian likelihoods such as that of quadratic losses). (2) To me, the proposed ensemble method is unconvincing: Why would you choose an artificially discretized prior? Doing so implies that the posterior derived with your algorithm shares the same support (by virtue of the KLD being infinity whenever prior and posterior have mismatched support). In particular, this means that if your countably many atoms that are supported by the prior do not feature any 'good' values, your posterior will never be any good, either. I don't understand why one would ever choose a prior of this kind if one could alternatively choose one that is absolutely continuous with respect to the Lebesgue measure on \Theta. (Even computationally this makes no sense to me, since variational inference is fairly scalable these days.)

Correctness: While I did not find any claims that were wrong, the presentation is fairly sloppy in more than a few places (see Clarity section). This makes it harder to assess the correctness of the claims than it should be and needs to be addressed.

Clarity: Overall, I would characterize the paper as providing an interesting contribution whose presentation is suboptimal. I would really like to see this paper accepted, but in order for me to defend this stance the following weaknesses in the presentation need to be addressed thoroughly: (1) Integration of prior literature into main text (see (A) -- (C) in next section) (2) It would be much better to state the assumption on the loss and prior in a separate assumption latex environment -- since the assumption of boundedness on p is made throughout, it would be appropriate to replace l. 81/82 on p. 3 with an actual explicit statement of the assumption. On that note, the text is not clear on whether the likelihood function is bounded in theta, x, or both. It would be necessary to be clear on this and write it out explicitly in a separate block of text. (This also resolves the awkward reference back to 'the assumptions stated in Section 3' that is made in Thm 2.) (3) p.5, l.193/194: Since this notion of diversity/disagreement is referred to throughout the paper (and is a very useful one in my mind), it would be worth elaborating on it in a separate paragraph: Why is it meaningful, what is the intuition behind V (in more detail)? (4) p.6, l. 233: Claiming that a sum of equally weighted dirac measures is a 'new prior' is exaggerated and incorrect. Instead, it would be more appropriate to simply state that "we restrict ourselves to a family of priors made up of equally weighted dirac measures." (5) What is the point of the discretized priors? Is there any tangible advantage over priors that are absolutely continuous with respect to the lebesgue measure? As pointed out above, I can see many drawbacks of discretizing the prior (and thereby the posterior) without seeing many advantages of the approach.

Relation to Prior Work: While the paper extensively relates to prior literature in a separate section that I found to be thorough and enjoyable, the paper would benefit from integrating some of the most important relations to prior work into the main text: (A) on p.2, l. 37 the authors claim that the paper shows 'that Bayesian model averaging provides suboptimal generalization performance' under misspecification. As the authors go on to correctly point out later though, this is not really a new finding (p.2, l. 56-59). It would be best to instead claim that the current paper *confirms* a well-known finding (the subpar generalization performance) with a new tool (namely the second-order PAC-inequality) that provides different insights into the mechanics of why one observes this phenomenon. (B) While the authors refer to the work in [28] a couple of times, I feel it would be best to point out this very natural connection when the generalized variational objectives are presented, i.e. in the pre- or post-test of eq. (4). (C) Similarly, the work of [24] is actually very close in spirit to what the authors propose to do: Rather than optimizing objectives with respect to the log-score, the authors optimize objectives with respect to the log predictive. This is exactly what equation (3) advocates, and so it makes sense to point out the prior literature that has sought to tackle the same/similar objectives. (I also recommend studying the references of [24], since I think that there is a related literature sometimes referred to as 'direct loss minimization' that is inspired by a similar observation as eq. (3).)

Reproducibility: Yes

Additional Feedback:


Review 2

Summary and Contributions: This work performs a novel theoretical analysis of the generalization capacity of Bayesian model averaging under model misspecification and provides strong theoretical arguments showing that Bayesian methods are suboptimal for learning predictive models when the model family is misspecified.

Strengths: These theoretical insights may help to better understand the generalization performance of Bayesian approaches and better explain the relationship between ensembles and Bayesian approaches.

Weaknesses: However, the current version is not very well justified and has limited novelty, and it’s not up to the high standard of NeurIPS. So, I am leaning towards a rejection for NeurIPS. The reasons for this decision are explained below.

Correctness: The technical content of the paper appears to be correct, I did not check it in detail.

Clarity: Some parts are not clear.

Relation to Prior Work: More discussion is needed.

Reproducibility: No

Additional Feedback: This work performs a novel theoretical analysis of the generalization capacity of Bayesian model averaging under model misspecification and provides strong theoretical arguments showing that Bayesian methods are suboptimal for learning predictive models when the model family is misspecified.Using novel second-order PAC-Bayes bounds, they derive a new family of Bayesian-like algorithms. These theoretical insights may help to better understand the generalization performance of Bayesian approaches and better explain the relationship between ensembles and Bayesian approaches. The technical content of the paper appears to be correct, I did not check it in detail. However, I found that there are several shortcomings: 1. I found the paper as a whole a little hard to follow. For a specific example of this see the line 53. 2. The methodologyis a bit confusing, and some parts are unclear. For example, the introduction and related works lack the logic of why this research was studied and the innovation of this paper. 3. There are some mistakes in the paper. Such as, line 19, “has”->”have”; line 23, “unrevolved” ->“unresolved”; line 56, there are two “that”; line 213, “contributions” ->“contribution”; line 315, “is” ->“are”; etc. 4. The experiments show good results on a few interesting cases. However, I think the empirical results are not thorough and convincing enough yet.The authors said their analysis was performed under unsupervised settings or density estimation, but this was not reflected in the experiments. 5. Regarding references, it is better notto make excessive citation to arXiv papers and cite large groups of papers without individually commenting on them, such as line 19, line 24, line 57, line 62, etc. Furthermore, more consistence is needed at a NIPS level, for example, line 441. However, the current version is not very well justified and has limited novelty, and it’s not up to the high standard of NeurIPS.


Review 3

Summary and Contributions: The paper analyzes the generalization performance of Bayesian model averaging under model misspecification. They focus on iid data and use second-order PAC-Bayes bounds.

Strengths: The paper studies a very important problem: when should Bayesian inference be preferred? when could it fail? They study Bayesian model averaging, which has been demonstrated in other works that produces good generalization performance. The paper shows that Bayesian model averaging provides suboptimal generalization performance when the model is misspecified. In particular, Bayesian posterior is the minimum of a first-order PAC-Bayes bound, which can be quite loose when the model family is misspecified. This naturally leads to their algorithms that use second-order PAC-Bayes bounds that are supposedly tighter. This analysis is very clear and intuitive. It is revealing about how existing Bayesian methods may fall short. And it also naturally leads to improved algorithms. The whole argument is really clean and nice. On the algorithms side, they also consider both variational inference and ensemble methods, illustrating the generality of the idea.

Weaknesses: I think two aspects of the empirical studies may be improved. One is the computational cost of the proposal. It is commonly seen that appealing to higher order bounds might significantly increase the computational cost. (this could be why existing methods have stayed with first order bounds.) However, the empirical evaluation of the proposal focus on deep neural networks and generalization performance performance. It would be interesting to see how much more computation is required to achieve the gain. A second aspect is the model. The current empirical studies focus on deep neural networks. While I appreciate that the authors focus on a complicated model that are more realistic, deep neural networks also involve many optimization complications together with model misspecification. It could be helpful also to illustrate the improvement of second order bounds in a simpler generative model and a simple type of model misspecification. Is there a simple model and a simple misspecification that can illustrate the improvement of second order bounds over first order bounds. This will help the readers build intuition over when second order bounds shall be used per limited computation.

Correctness: The paper appears correct.

Clarity: The paper is quite well-written.

Relation to Prior Work: The paper adequately discusses prior work.

Reproducibility: Yes

Additional Feedback: See above. -------------- Thank you to the authors for the rebuttal. I have read the rebuttal and my evaluation stays the same.


Review 4

Summary and Contributions: This manuscript provides a theoretical analysis of generalization performance from the perspective of PAC-Bayes methods, focusing on the case of misspecified models. Generalization here is quantified by the cross-entropy of the posterior predictive distribution with respect to the true (unknown) data-generating distribution. Within this framework, the authors show why Bayesian posteriors provide suboptimal generalization performance when the true model does not belong to the model set. In short, Bayesian posteriors emerge from the minimization of a first-order (linear) bound; thus, solutions will always at the borders of the model space (with the posterior collapsing to a point). Instead, the authors propose second-order PAC-Bayes bounds, based on second-order extensions of Jensen's inequality. These second-order bounds introduce a correction based on a variance term, with the effect of promoting "diversity" between predictions of the model. The distributions obtained using these bounds obtain better generalization (i.e., lower cross-entropy) than Bayesian posteriors. The authors also propose similar second-order bounds applied to ensemble models, and optimization algorithms to minimize these bounds. Finally, the theoretical results are validated with a number of empirical studies (e.g., on Fashion-MNIST and CIFAR-10). ====== After rebuttal ========= I was happy with the authors' response and I confirm my highly positive vote.

Strengths: Soundness: This paper is very well-grounded in theory and mathematical proofs, all available in the lengthy appendix. There is also quite an extensive empirical examination of properties of the proposed methods on toy problems in the Appendix. Significance and novelty: This work is novel and timely. Its significance is potentially very high, as uncertainty quantification and generalization are still largely an open problem with many open theoretical questions in machine learning. The proposed framework of second-order PAC bounds seems like a very promising and interesting research direction. Relevance: This work is of very high relevance to the NeurIPS audience, by providing theoretical explanations for common intuition (e.g., why ensemble diversity should be promoted in the first place), and techniques to potentially improve generalization.

Weaknesses: Soundness: The empirical evaluation of the method, while acceptable for this work (which is mostly theoretical, and it's not necessarily about "beating SOTA"), however seems to miss a direct comparison with fully Bayesian approaches, which I would have expected considered how good part of the work, after all, is to show how the Bayesian posterior can be inadequate for generalization. In Section 7 (and Figure 3), instead, the authors compare their approach against some simple baselines - a MAP (point estimate) or mean-field variational solution (VAR), where the latter is a coarse approximation of a Bayesian posterior. While I appreciate that computing the full Bayesian posterior might be infeasible (it is an open problem after all), the authors could consider adding some strong baseline for posterior approximation, such as state-of-the-art forms of approximate posterior sampling, e.g. SWAG (Maddox et al., NeurIPS 2019). As is, the current empirical results on non-toy datasets are promising but not outstanding.

Correctness: As far as I can tell, the theoretical claims are correct, although I did not fully verify all proofs in the Appendix in detail. The empirical claims are supported from the data, with the caveat mentioned above of lacking of a strong baseline for Bayesian inference.

Clarity: The paper is well-written and generally quite easy to follow, even in its more mathematical parts.

Relation to Prior Work: Prior work and differences are discussed quite well.

Reproducibility: Yes

Additional Feedback: A few typos: line 23: unrevolved --> unresolved line 244: is provide --> is provided line 484 (appendix): we first proof --> we first prove line 589: numerlically --> numerically

[Author Response · NeurIPS 2020]

We thank all the reviewers for their insightful comments and suggestions.

**Reviewer 1:**

*(1) Prior Literature (A)-(C):* Thanks for the suggestions. We agree and will integrate them into the paper.

*(2) Separate assumption latex environment:* We fully agree. Thanks for the suggestion.

*(3) Diversity*: This is a good point. We will add a paragraph.

*(4) Prior*: It was not our intention to claim that this is a **novel** prior family. We will rewrite that.

*(5) Discretized Prior*: We fully agree with you that there are many arguments against the use of a discretized prior. It is
not a natural choice. We do not advocate for this kind of priors, we use them because they allow us to characterize
ensembles as variational methods which approximate the posterior distribution with a finite set of particles (instead
of a continuous density belonging to some parametric family, as it is usually the case). One could, for example, use
a mixture of Gaussians, which (in my opinion) will be more powerful and will also capture multimodality, but their
treatment will be much more involved. But this characterization, which has been previously mentioned in [53], helps us
to establish a direct link between our theoretical analysis and ensemble learning algorithms. So, the main purpose of
Section 6.3 (and the use of the discretized prior) is to show that this work provides a candidate theoretical tool to study
the generalization capacity of ensembles-like algorithms. In any case, we will add a caveat about the use of these priors
and the reasons because we introduce them.

*Bernstein/Hoeffding type assumptions:* Theorem 1 was initially presented in [2] and was restricted to "losses/likelihoods
(and priors) under which the usual Bernstein/Hoeffding type assumptions hold". But later, [17,45] showed that Theorem
1 was also valid under general unbounded losses.

**Reviewer 2:**

Thanks for enumerating the typos. We plan to send the paper to a native English speaker.

The paper focuses on density estimation because it is easier to present (and the notation is a bit simpler), but it readily
applies to conditional densities too. Experiments were performed in supervised problems involving Bayesian neural
networks because these are the settings where the community is currently paying more attention.

**Reviewer 3:**

*Computational Complexity:* Thanks for the point. You are right and we haven't discussed it in the paper. But we are
glad to introduce the following discussion either in the main body or in the appendix. From our theoretical analysis, we
derive a new **loss** function, introduced in Equation (4). The computation of the gradient of this loss function is
more complex than the standard variational loss function (see Equation (2)), but not too much. Looking at Equation
(C.12), we can see that it is a constant factor 2 more complex. However, it is not clear if the optimization of this new
loss function is or not more involved than the optimization of the standard variational loss function (Equation (2)), this
is going to depend on the loss landscape and the noise of the gradients. More research in this direction is needed in
order to properly answer this question.

*Evaluation Simpler Models:* This is also a good point. This is mostly a theoretical paper, but we decided to illustrate
this approach in a *modern* or *hot* problem to highlight that this theoretical analysis could be potentially very useful
(and it is not just a theoretical curiosity). In any case, Figures C5-C9 in the appendix further illustrate this approach
on simpler models. We are currently working on an evaluation based on a mixture of Gaussians where the number of
components is misspecified, and we plan to include it in the appendix (if the paper is accepted). And, of course, future
works will focus on extensive empirical evaluations of this approach.

**Reviewer 4:**

*Comparisions with stronger baselines:* Thanks for the reference, we were not aware of SWAG, and we will try to
include it in the experimental evaluation, we think it is a fair comparison. We use standard mean-field variational
because is simple and widely used in this kind of problems. In this case, the mean-field approach defines the solution
space. So, the variational approach gets the minimum of Equation (2) within this solution space. We think it is fair
to compare with the minimum of Equation (4) within the same solution space. Because, in this case, we can clearly
evaluate which is the effect of considering the new presented loss function. Richer solution spaces could be considered
too (e.g. "The k-tied Normal Distribution" Swiatkowski et al (2020)). But the fair comparison will be to evaluate both
the minimum of Equation (2) versus the minimum of Equation (4) within the same solution space. Moreover, one could
eventually think if a counter-part version of SWAG could be defined using the insights derived from this paper. In any
case, we agree that an extensive empirical evaluation of this approach is needed to see whether this research direction is
able to provide new SOTA results.

[Meta-Review · NeurIPS 2020]

In this paper, the authors analyze the problem of poor performance of Bayesian model averaging under model misspecification using new second-order PAC-Bayes bounds. And they use their tools to suggest an alternative model to handle misspecification. Three of four reviewers are excited about the novelty of the approach and its important practical contribution. That being said, the authors need to be careful to follow up on their promises (in the author feedback) for clarification throughout the paper. And the authors need to make sure to correct the error in the author feedback flagged by Reviewer 1 (please see revised review).